# Dopamine Spraying Protects Against Cadmium-Induced Oxidative Stress and Stimulates Photosynthesis in Soybean Plants

**DOI:** 10.3390/plants14223411

**Published:** 2025-11-07

**Authors:** Andreza Sousa Carmo, Caio Victor da Silva Pontes, Caroline Cristine Augusto, Bruno Lemos Batista, Allan Klynger da Silva Lobato

**Affiliations:** 1Núcleo de Pesquisa Vegetal Básica e Aplicada, Universidade Federal Rural da Amazônia, Paragominas 68627-451, PA, Brazil; andreza.carmo@discente.ufra.edu.br (A.S.C.); agrocaiopontes@gmail.com (C.V.d.S.P.); 2Centro de Ciências Naturais e Humanas, Universidade Federal do ABC, Santo André 09280-560, SP, Brazil; caroline.c.augusto@gmail.com (C.C.A.); bruno.lemos@ufabc.edu.br (B.L.B.)

**Keywords:** carbon fixation, *Glycine max*, heavy metal, neurotransmitter, stress indicators

## Abstract

Cadmium (Cd) is a non-essential element that induces reactive oxygen species (ROS) production and damages the photosynthetic apparatus. Dopamine (DOP) is a neurotransmitter that plays a role in metabolism as an antioxidant. This research aimed to investigate whether exogenous DOP mitigates Cd-induced oxidative stress in soybean by assessing antioxidant metabolism, stress indicators, nutritional status, pigments, chlorophyll fluorescence, gas exchange, and biomass. The experiment was randomized with four treatments: two with Cd concentrations (0 and 500 µM Cd, described as—Cd and +Cd, respectively) and two DOP levels (0 and 100 µM DOP described as—DOP and +DOP, respectively). DOP mitigated Cd-induced damage by enhancing the antioxidant system and protecting the photosynthetic apparatus. This neurotransmitter positively modulated the enzymes superoxide dismutase (38%), catalase (27%), ascorbate peroxidase (23%), and peroxidase (31%), alleviating Cd-induced oxidative stress. In addition, DOP promoted increases in the effective quantum yield of PSII photochemistry (26%), photochemical quenching coefficient (18%), and electron transport rate (26%). Simultaneously, the neurotransmitter stimulated increases in the net photosynthetic rate (29%), stomatal conductance (35%), water use efficiency (38%), and instantaneous carboxylation efficiency (39%). Our results indicate that DOP exogenous increases tolerance to Cd-induced stress in soybean plants.

## 1. Introduction

Soybean [*Glycine max* (L.) Merrill] is one of the most important crops in the world due to the high protein contents, oil, and carbohydrates present in seeds and grains, which makes cultivation essential for food production, animal feed, and industrial products [1,2]. Brazil is the world’s largest crop producer, followed by the USA and Argentina [3]. In the 2023–2024 harvest, approximately 46.1 thousand hectares of Brazilian territory were allocated to legume cultivation, producing 147.7 thousand tons [4]. However, this crop is frequently exposed to environmental stresses in the central producing regions worldwide that compromise yield [5], such as soil contamination by heavy metals in areas with high agricultural and mining activity [6]. In general, soybean plants exhibit low tolerance to heavy metal stress, which severely affects plant performance [7].

Cadmium (Cd) is a non-essential metallic element for plant growth, phytotoxic even in low concentrations, being harmful to human and animal health and causing deleterious effects in agriculture [8,9]. It has high solubility in the soil and is easily absorbed by plant roots and transported to shoots [10]. The root, the main organ for nutrient uptake, is also the first tissue affected by Cd stress, which reduces its growth and compromises the plant’s nutritional status [11,12]. This effect arises from ionic competition at the root surface [13], where Cd^2+^ competes with essential elements, particularly divalent cations, for transport channels, leading to disturbances in nutrient assimilation and in the specific functions of these elements [14].

Cd-stressed plants exhibit physiological, biochemical, and molecular changes [15]. At the biochemical level, this metal stimulates the synthesis of reactive oxygen species (ROS), causing damage to the permeability and functions of cells [16,17,18]. In response to the overproduction of ROS, plants developed a redox detoxification system, based on antioxidant defense mechanisms by increasing the superoxide dismutase (SOD), peroxidase (POX), catalase (CAT), and ascorbate peroxidase (APX) activities, which act to preserve homeostasis at the cellular level [19]. In a physiological context, Cd toxicity can negatively interfere with the photosynthetic system, causing damage to the light collection complex, photosystems I and II, and enzyme activities involved in the Calvin cycle [20,21]. In general, oxidative stress causes damage to chloroplast structures, reduces stomatal conductance, interrupts chlorophyll synthesis, and consequently reduces the photosynthetic rate [22], resulting in a decrease in biomass accumulation and affecting plant growth and survival [22].

Dopamine (DOP), like norepinephrine and epinephrine, belongs to the group of catecholamines and is considered a biogenic amine [23]. Catecholamines play an essential role as neuromodulators or neurotransmitters in plants, animals, and humans [24]. DOP, a water-soluble antioxidant, is a secondary metabolite with high antioxidant capacity that enhances the adaptability of plants to stresses [25]. Exogenous application of DOP has been reported in recent studies to be efficient in alleviating biotic and abiotic stresses, including drought [26], salinity [27], nutritional deficiency [28], alkalinity [29], heavy metals [30], waterlogging [31], cold [32], organic pollutants [33] and diseases [34]. It frequently enhances stress tolerance by positively regulating nutrient uptake, transport, and assimilation [35], improving gas exchange, photosynthetic pigments, and chlorophyll fluorescence parameters [36], as well as boosting antioxidant enzyme activities, reducing ROS accumulation [37], and protecting cells from lipid peroxidation [38].

It is hypothesized that Cd excess induces the overproduction of ROS, thereby interfering with the functioning of the photosynthetic machinery [39]. However, the DOP molecule plays an antioxidant role in metabolism, representing an interesting opportunity to prevent oxidative stress in plants [40,41,42]. Considering that the literature does not provide results on DOP roles in soybean plants under Cd stress and recurrent environmental contamination by heavy metals, this study aimed to investigate whether the exogenous application of DOP can mitigate oxidative stress and the repercussions on the photosynthetic apparatus in soybean plants under Cd intoxication, evaluating responses associated with antioxidant metabolism, stress indicators, nutritional status, photosynthetic pigments, chlorophyll fluorescence, gas exchange, and biomass.

## 2. Results

### 2.1. DOP Minimized Cd Concentrations in Plants Exposed to Toxicity

Plants treated with 500 µM Cd^2+^ showed increased Cd contents in their tissues (Table 1). However, plants subjected to treatment with 500 μM Cd^2+^ associated with 100 μM DOP presented significant reductions in contents of this metal by 60%, 29%, and 38% in the root, stem, and leaf, respectively, when compared to the treatment with 500 μM Cd^2+^.

### 2.2. Neurotransmitter Positively Regulates Nutrient Contents

Cd excess caused reductions in nutrient contents in the root, stem, and leaf (Table 2). However, plants subjected to treatment with DOP and Cd^2+^ increased potassium (K), calcium (Ca), magnesium (Mg), manganese (Mn), iron (Fe), and zinc (Zn) contents by 34%, 37%, 22%, 58%, 57% and 26% (root), 19%, 24%, 51%, 37%, 35% and 18% (stem), and 20%, 16%, 25%, 23%, 22% and 12% (leaf), in this order, compared to plants treated with Cd alone.

### 2.3. DOP Alleviated the Cd Impacts on Photosynthetic Apparatus

Cd excess caused reductions in photosynthetic pigments (Table 3). However, treatment with 100 µM DOP in plants stressed by Cd^2+^ promoted significant increases of 37%, 45%, 39%, and 45% in chlorophyll a (Chl *a*), chlorophyll b (Chl *b*), total chlorophyll (Total Chl), and Carotenoids (Car), respectively, compared to plants treated only with Cd. Regarding chlorophyll fluorescence, Cd excess promoted reductions (*p* < 0.05) in the values of maximal fluorescence yield of the dark-adapted state (F_m_), variable fluorescence (F_v_), and maximal quantum yield of PSII photochemistry (F_v_/F_m_) (Figure 1), while minimal fluorescence yield of the dark-adapted state (F_0_) did not show significant changes. However, the combination of DOP and Cd^2+^ occasioned increases of 8%, 12%, and 3% in F_m_, F_v_, and F_v_/F_m_, in this order, and a 3% reduction in F_0_ about treatment with Cd alone. Furthermore, plants treated with Cd suffered reductions in effective quantum yield of PSII photochemistry (Φ_PSII_), photochemical quenching (qP), and electron transport rate (ETR) and increases in nonphotochemical quenching (NPQ), relative energy excess at the PSII level (EXC), and ratio between the apparent electron transport rate and net photosynthetic rate (ETR/*P*_N_) (Table 3). However, Dopamine-treated plants exposed to excess Cd promoted increases of 26%, 18%, and 26% in Φ_PSII_, q_P_, and ETR, respectively, and reductions of 6%, 13%, and 2% in NPQ, EXC, and ETR/*P*_N_, respectively, compared with plants exposed to the same Cd treatment without dopamine. For gas exchange, plants subjected to excess Cd showed reductions (*p* < 0.05) in net photosynthetic rate (*P*_N_), stomatal conductance (*g*_s_), water-use efficiency (WUE), and instantaneous carboxylation efficiency (*P*_N_/*C*_i_) and increases in transpiration rate (*E*) and intercellular CO_2_ concentration (*C*_i_) (Table 3). However, when plants under Cd^2+^ stress were sprayed with DOP, there were increases of 29%, 35%, 38%, and 39% in *P*_N_, *g*_s_, WUE, and *P*_N_/*C*_i_, respectively, and decreases in both variables (*E* and *C*_i_) were 6%, in the treatment with 500 μM Cd^2+^.

### 2.4. Benefits in Antioxidant Defense and Lower ROS Concentrations Induced by DOP

The activities of antioxidant enzymes (SOD, CAT, APX, and POX) had significant increases when subjected to Cd stress (Figure 2). However, treatment with 100 µM DOP combined with 500 µM Cd^2+^ resulted in increases (*p* < 0.05) in SOD, CAT, APX and POX activities of 38%, 27%, 23% and 31%, respectively, compared to the same treatment in the absence of DOP. Regarding stress indicators, excess Cd stimulated the ROS accumulation (Figure 3). However, spraying DOP on plants stressed with Cd caused significant reductions in the levels of oxidative compounds superoxide (O_2_^−^), hydrogen peroxide (H_2_O_2_), malondialdehyde (MDA), and electrolyte leakage (EL) of 28%, 14%, 23% and 6%, respectively, when compared to plants treated with Cd alone.

### 2.5. DOP Mitigated the Impacts Caused by Cd Excess on Biomass

Biomass was significantly reduced in plants exposed to Cd excess (Figure 4 and Figure 5). However, DOP sprayed in plants stressed with Cd^2+^ induced increases in leaf dry matter (LDM), root dry matter (RDM), stem dry matter (SDM), and total dry matter (TDM) of 89%, 67%, 98%, and 89%, respectively, compared to plants exposed only to Cd excess.

## 3. Discussion

Cd concentrations detected in soybean plants’ leaves, stems, and roots demonstrate the toxicity induced by the exogenous application of 500 μM Cd. On the other hand, spraying 100 μM DOP induced reductions in Cd contents, highlighting the role of this neurotransmitter and attenuating the toxic effects caused by this heavy metal. Dopamine alleviates Cd-induced stress by regulating the expression of genes involved in metal uptake and detoxification. In particular, overexpression of the MdTyDC gene, associated with neurotransmitter synthesis, helps reduce Cd^2+^ transport to the shoots [43]. Similarly, the application of 100 μM DOP in *Malus hupehensis* mitigated Cd stress, reducing the concentration and accumulation of this metal [44]. Cd overaccumulation in the root is a mechanism of plant tolerance to toxicity. Often, plants exposed to Cd excess restrict the metal in the roots, reducing its translocation to the shoot of the plants [45,46], corroborating our results with higher concentrations in root tissue. High Cd levels in *Eucalyptus urophylla* plants promoted immobilization of this metal in the roots, reducing Cd levels in leaves and stems [47].

Plants exposed to Cd stress showed an imbalance in the homeostasis of essential elements. However, DOP mitigated the adverse effects of Cd and promoted increased contents of macronutrients (K, Ca, and Mg) and micronutrients (Mn, Fe, and Zn) found in tissues. This improvement in nutrient uptake may be associated with the observed increases in RDM. The roots absorb essential nutrients, making this tissue the first organ to perceive environmental restrictions. Additionally, successful nutrient acquisition is determined by the architecture of the root system and root growth rate [48,49]. Cd^2+^, as a homolog of metallic elements, can affect the absorption of essential nutrients, especially divalent cations such as Ca^2+^, Mg^2+^, Mn^2+^, Fe^2+^, and Zn^2+^, due to their similar chemical properties, interfering with element transport pathways and membrane permeability [50,51]. Cd^2+^ uptake occurs through transporters of essential mono and divalent metals, competing with cations such as K^+^, Ca^2+^, Mg^2+^, Mn^2+^, Fe^2+^, and Zn^2+^, as well as through ion transport channels, thereby reducing the uptake of essential elements by the roots [8,50,52]. Three ordinary transporter families have been described to be involved in Cd uptake: NRAMP (natural resistance-associated macrophage protein), ZIP (zinc/iron-regulated transporter-like protein), and YSL proteins (yellow stripe-like 1) [53]. *Malus hupehensis* seedlings treated with DOP and subjected to stress due to nutrient deficiency, observed that DOP supplementation alleviated the stress induced by the deficiency, improving the nutritional status of the plants evaluated [54]. Similarly to our results, application of 100 μM DOP can alleviate stress due to low N supply by regulating the absorption of essential nutrients [55].

Plants treated with DOP and exposed to Cd showed increases in enzymatic activity. The increases observed in the activities of SOD, CAT, APX, and POX suggest benefits associated with prior treatment with neurotransmitters. DOP can directly mitigate the damage caused by ROS, as it is a water-soluble antioxidant that acts during the synthesis of melanin, a product linked to DOP oxidation, and is an effective scavenger of free radicals or through antioxidant enzymes [29,56]. Upregulation in the expression of antioxidant genes such as ascorbate peroxidase (cAPX), monodehydroascorbate reductase (*MDHAR*), dehydroascorbate reductase (*DHAR*), and glutathione reductase (*cGR*) may also contribute to the antioxidant capacity of the neurotransmitter [42]. The exogenous application of DOP promoted an increase in the activities of POX, CAT, and SOD, compared to watermelon seedlings exposed to cold stress [57]. Similarly, in response to bisphenol A stress, dopamine enhanced the activities of antioxidant enzymes in cucumber plants [33].

Plants exposed to excess Cd promoted the overaccumulation of ROS. However, applying DOP attenuated the adverse effects of stress indicators (O_2_^−^, H_2_O_2_, MDA, and EL). DOP and melanin, the products of their oxidation, play a direct role in ROS elimination, thereby preserving high antioxidant capacity [30,58]. Antioxidant enzymes convert ROS into non-toxic molecules, preventing cell disintegration by ROS [59]. For example, SOD is involved in the conversion of O_2_^−^ into H_2_O_2_, while CAT degrades H_2_O_2_ into water and oxygen molecules. Furthermore, POX and APX decompose lipid peroxides and detoxify H_2_O_2_ through the ascorbate-glutathione cycle [60,61]. Exogenous treatment using 100 μM DOP in lettuce plants subjected to nitrogen deficiency mitigated oxidative stress, stimulating the antioxidant defense system [62]. DOP supplementation inhibited ROS accumulation, stimulated antioxidant enzymes, and protected plant cells from lipid peroxidation in soybean plants under salinity conditions [27]. These results corroborate our observations, demonstrating that the increase in antioxidant enzyme activities induced by DOP attenuates the oxidative stress caused by Cd.

Photosynthetic pigments were stimulated by spraying plants with DOP under Cd stress, as these increases were related to the improvement in nutritional status and significant increases in the contents of Mg and Fe. DOP increased the Mg and Fe contents and reduced the deleterious effects on pigments induced by Cd excess, improving the availability of these elements for the synthesis of Chl *a*, Chl *b*, Total Chl, and Car. Cd stress negatively interferes with physiological processes; the main sites of action of this metal are the photosynthetic system, its pigments, and carotenoid [63], having an effective role in increasing the activity of the chlorophyll degradation enzyme (chlorophyllase) or in suppressing the expression levels of key genes for chlorophyll synthesis [64]. Furthermore, Fe helps to improve chlorophyll contents and the synthesis of other pigments that are directly involved in collecting light for photosynthesis; its deficiency interferes with chloroplast component proteins, such as the cytochrome *b6f* complex (Cyt b6f) and ferredoxin (Fd), inhibiting electron transport and the efficiency of photosystem II [65,66]. Mg is the central element of the chlorophyll molecule and the activation of the enzyme ribulose-1,5-bisphosphate carboxylase-oxygenase (RuBisCo), being a fundamental nutrient for the photosynthesis process [67,68]. Corroborating with our results, the application of 100 μM DOP increased the Chl *a*, Chl *b*, and Total Chl contents by 43%, 65%, and 50%, respectively, in tomato seedlings under saline stress [69]. DOP treatment mitigated cold stress in banana plants by enhancing chlorophyll accumulation, which was associated with the downregulation of chlorophyll degradation–related genes (*MDC*, *PPH*, *PaO*, and *RCCR*), suggesting that DOP prevents chlorophyll breakdown [32].

The application of DOP attenuated the toxic effects caused by excess Cd on chlorophyll fluorescence. The reduction in the F_0_ value and the increase in F_v_, F_m_, and F_v_/F_m_ indicate improvements induced by the neurotransmitter in the PSII reaction center, enhancing its ability to collect and convert light energy. Cd stress impairs the absorption and conversion capacity of pigment molecules within PSII reaction centers [70]. On the other hand, DOP enhances the activity and stability of photosystems, particularly PSII, and also increases the concentration of antenna proteins [71]. Our results demonstrate that the changes found in F_v_ originate from F_m_. In other words, Cd stress inhibited Fm. Fm is a key parameter in chlorophyll fluorescence in plants, being the highest level of fluorescence emitted by chlorophyll in a leaf after a plant is subjected to a dark-adapted state. The difference between Fm and F_0_ yields the F_v_, which is then used in the ratio F_v_/F_m_, a standard indicator of the plant’s photosynthetic efficiency.

Decreases in F_v_/F_m_ were observed in common bean plants exposed to Cd [72]. Cd contamination in sunflower plants promoted reductions in F_m_ and F_v_/F_m_, while there was an increase in F_0_ [73]. In other hand, *Malus hupehensis* leaves treated with 100 μM DOP showed an increase in F_v_/F_m_ [74].

Cd^2+^ excess caused reductions in Φ_PSII_, q_p,_ and ETR. On the other hand, 100 μM DOP mitigated the harmful interference caused by Cd. Cd affects metabolic and physiological processes, such as photosynthesis and photochemical reactions, with PSII being particularly sensitive. In PSII, Cd interferes with the oxygen evolution cycle at the donor site, inhibiting the transfer of electrons from Quinone A to Quinone B at the electron acceptor site [75]. Furthermore, DOP caused a decrease in NPQ, EXC, and ETR/PN values, confirming that exogenous DOP promotes improvements in the capture and utilization of light energy, as well as reducing the heat dissipation of light energy [36].

The reduction in q_p_ and ETR values, together with the increase in NPQ and the decline in PN, indicates that Cd toxicity may delay plant growth by promoting the formation of ROS, which damage the photosynthetic apparatus and impair the efficiency of electron transfer in the light reaction center [76]. Application of 100 μM DOP increased Fv/Fm, ΦPSII, and qP, while reducing NPQ in cucumber leaves under mildew stress, indicating that exogenous DOP enhances the efficiency of light energy absorption and conversion [77]. Additionally, the role of the *TyDC* gene in DOP biosynthesis in apple plants and its contribution to drought stress response resulted in increases in F_v_/F_m_, Φ_PSII_, and q_p_, as well as a reduction in NPQ in transgenic lines (overexpressing *MdTyDC*) after 40 days of stress. These effects were associated with the higher DOP accumulation derived from *MdTyDC* overexpression [78].

Pre-treatment with 100 μM DOP in plants stressed by Cd promoted increases in *P*_N_, *g*_s_, WUE, and *P*_N_/*C*_i_, and reducing in *E* and *C*_i_. Cd toxicity damages the structures of chloroplasts and thylakoids, causing disturbances in the PSII reaction center, destruction of the oxygen-evolving complex (OEC), and blockage of electron transfer from Q_A_ to Q_B_. These effects inhibit energy transfer and photochemical activity, resulting in a reduction in the photosynthetic rate [52,79,80]. Increases in *P*_N_, *g*_s_, WUE, and *P*_N_/*C*_i_ induced by DOP spray are related to the increases obtained in Φ_PSII_, q_p,_ and ETR, indicating that this molecule provides benefits in the photosynthetic apparatus, effectively improving photochemical efficiency, RuBisCo activity, and promoting carbon assimilation [71,81]. The reduction in C_i_ and increase in *g*_s_ suggest better stomatal performance, maximizing CO_2_ assimilation [82]_._ The transpiration process is intrinsically linked to water absorption and stomatal opening of plants [83], as detected in this study by the increase in WUE and reduction in *E*. Cowpea plants exposed to Cd exhibited reductions in *P*_N_, *g*_s_, *E*, WUE, and *P*_N_/*C*_i_, accompanied by an increase in *C*_i_ [84]. Pre-treatment with different concentrations of DOP (0, 50, 100, 150, and 200 μM) in cucumber plants exposed to nitrate-induced stress (50 or 500 μM) markedly increased *P*_N_, gs, *E*, and *C*_i_ following DOP application [36].

Cd stress reduced biomass. However, DOP mitigated the deleterious effects induced by the metal on leaves, stems, and roots. The harmful impact of biomass reduction provoked by Cd is intrinsically related to essential nutrient uptake, chlorophyll synthesis, photosynthesis, gas exchange, membrane permeability, and ROS overproduction [85,86,87]. On the other hand, the values of LDM, SDM, RDM, and TDM revealed relevant increases when plants were pre-treated with DOP, being related to the roles played by this neurotransmitter in nutritional status, antioxidant system, and ROS elimination, photosynthetic pigments, chlorophyll fluorescence, and gas exchange described in this research. Cowpea plants treated with 500 μM Cd^2+^ showed reductions in LDM, RDM, SDM, and TDM [88]. Corroborating this research, *Brassica oleracea* seedlings treated with DOP (0, 50, 100, and 200 µM) and exposed to hydrocarbon-induced stress exhibited increases in both shoot and root biomass [89].

## 4. Materials and Methods

### 4.1. Geographical Location and Growth Parameters

The experiment was conducted in the Paragominas Campus of the Federal Rural University of Amazonia, Paragominas, Brazil (2°55′ S, 47°34′ W). The research was performed in a greenhouse with regulated temperature and humidity. The minimum, maximum, and median temperatures were 25.2 °C, 31.4 °C, and 26.7 °C, respectively. The relative humidity throughout the trial period fluctuated between 60% and 80%.

### 4.2. Plants, Containers, and Acclimation

Seeds of *Glycine max* (L.) Merr. var. M8644IPRO Monsoy™ were germinated and grown in 1.2-L pots [90]. For plant nutrition, a nutrient solution with an initial ionic strength of 50% was used (4th day) and later adjusted to 100% after 2 days (6th day) [91]. After this period, the nutritive solution remained at total ionic strength.

### 4.3. Experimental Design, Plant Nutrition and Cd Excess

The experiment was randomized with four treatments: two with cadmium concentrations (0 and 500 µM Cd, described as—Cd and +Cd, respectively) and two concentrations of dopamine (0 and 100 µM DOP described as—DOP and +DOP, respectively). Five replicates for each of the four treatments were conducted, yielding 20 experimental units in the experiment, with one plant in each experimental unit. DOP concentrations were chosen based on previous research [37]. Cd treatments were defined in preliminary tests and results available in literature [88].

Plants received the following macro- and micronutrients supplied in the nutrient solution as described previously [91]. To simulate Cd^2+^ exposure, cadmium chloride (CdCl_2_) was used at concentrations of 0 and 500 μM Cd and applied over 10 days (day 20–30 after the start of the experiment), with a 5-day interval between applications. During the study, the nutrient solutions were changed at 07:00 h at 3-day intervals, with the pH adjusted to 6.5 using HCl or NaOH. On day 30 of the experiment, all plants’ physiological and morphological parameters were measured, and the leaf tissues were collected for nutritional and biochemical analyses.

### 4.4. Dopamine (DOP) Preparation and Application

12-day-old plants were sprayed with dopamine (DOP) or Milli-Q water (containing a proportion of ethanol equally used to prepare the DOP solution), spraying 10 mL per plant at 5-day intervals until day 30. The DOP solution (100 µM; Sigma–Aldrich, Saint Louis, MO, USA) was prepared as described previously [29].

### 4.5. Chlorophyll Fluorescence and Gaseous Exchange

Chlorophyll fluorescence was assessed using a modulated chlorophyll fluorometer (model OS5p; Opti-Sciences, Hudson, NH, USA), being measured in fully expanded leaves under light. Preliminary tests determined that the acropetal third of leaves in the middle third of the plant and that adapted to the dark for 30 min yielded the greatest F_v_/F_m_ ratio. Therefore, this part of the plant was used for measurements. The intensity and duration of the saturation light pulse were 7500 µmol m^−2^ s^−1^ and 0.7 s, respectively [92]. Gas exchange was measured using an infrared gas analyzer (LCPro+; ADC BioScientific, Hoddesdon, UK), being evaluated in all plants under a constant CO_2_ concentration (360 μmol mol^−1^ CO_2_), photosynthetically active radiation (800 μmol photons m^−2^ s^−1^), air-flow rate (300 µmol s^−1^), and temperature (28 °C) in the test chamber between 10:00 and 12:00 h [91].

### 4.6. Assessment of Antioxidant Enzymes, Soluble Proteins, and Stress Markers

Antioxidant enzymes (SOD, CAT, APX, and POX) and soluble proteins were isolated from leaf tissues. The extraction mixture was prepared by homogenizing 500 mg of fresh plant material in 5 mL of extraction buffer, which consisted of 50 mM phosphate buffer (pH 7.6), 1.0 mM ascorbate, and 1.0 mM EDTA. Samples were centrifuged at 14,000× *g* for 4 min at 3 °C, and the supernatant was collected [93]. Quantification of total soluble proteins was conducted [94]. SOD assay was performed at 560 nm [95], and SOD activity was quantified as units per mg of protein. CAT assay was measured at 240 nm [96], with CAT activity quantified as μmol H_2_O_2_ mg^−1^ protein min^−1^. The APX experiment was conducted at 290 nm [97], with APX activity quantified as μmol AsA mg^−1^ protein min^−1^. The POX assay was measured at 470 nm [98], with activity quantified as μmol tetraguaiacol mg^−1^ protein min^−1^. The O_2_ concentration was quantified at 530 nm [99]. Stress markers were isolated, specifically H_2_O_2_ and MDA [100]. The H_2_O_2_ concentration was quantified [101]. The MDA concentration was calculated utilizing an attenuation value of 155 mM^−1^ cm^−1^ [102]. EL was assessed using the methodology outlined in [103] and calculated as EL (%) = (EC_1_/EC_2_) × 100.

### 4.7. Assessment of Photosynthetic Pigments, Nutritional Composition, and Biomass

Chlorophyll and carotenoid concentrations were assessed using 40 mg of foliar tissue. The samples were homogenized in darkness using 8 mL of 90% methanol (Sigma-Aldrich™). The homogenate underwent centrifugation at 6000× *g* for 10 min at 5 °C. The supernatant was discarded, and the concentrations of Chl *a*, Chl *b*, Car, and total Chl were measured using a spectrophotometer (model UV-M51; Bel Photonics, Monza, Italy) following the protocol established in [104]. Samples (100 mg) of root, stem, and leaf tissues were pre-digested in 50 mL conical tubes using 2 mL of sub-boiled HNO_3_. Subsequently, 8 mL of a solution comprising 4 mL of H_2_O_2_ (30% *v*/*v*) and 4 mL of ultra-pure water was added and transferred to a Teflon digestion tube [105]. Cd, K, Ca, Mg, Mn, Fe, and Zn were quantified using an inductively coupled plasma mass spectrometer (model ICP-MS 7900; Agilent, Santa Clara, CA, USA). The biomass of root, stem and leaves were quantified by constant dry weights (g) following desiccation in a forced-air oven at 65 °C.

### 4.8. Data Analysis

The normality of the residuals was assessed using the Shapiro–Wilk test. The data underwent one-way ANOVA, and significant mean differences were assessed using the Scott–Knott test at a 5% probability level [106]. All statistical analyses employed R™ software version 4.3.3 [107].

## 5. Conclusions

Dopamine (DOP) mitigated the deleterious effects caused by Cd toxicity, regulating the antioxidant mechanism and protecting the photosynthetic apparatus of soybean plants. Exogenous DOP reduced oxidative damage by enhancing the activities of antioxidant enzymes, including superoxide dismutase, catalase, ascorbate peroxidase, and peroxidase, which detoxify reactive compounds such as hydrogen peroxide, superoxide, malondialdehyde, and electrolytes. Consequently, DOP alleviated negative effects on the photosynthetic apparatus, improving chlorophyll content, photosynthetic activity, PSII efficiency, and gas exchange in plants under Cd stress. In addition to the diverse roles of DOP in the antioxidant system, gas exchange, and photochemical efficiency of photosystem II, the neurotransmitter also improved the nutritional status of plants, contributing to the increase in biomass accumulation. Therefore, our results indicate that exogenous DOP increased tolerance to Cd-induced stress in soybean plants, but future studies with other soybean varieties and varying DOP concentrations, combined with transcriptomic or proteomic validation, are necessary.

## Figures and Tables

**Figure 1 plants-14-03411-f001:**
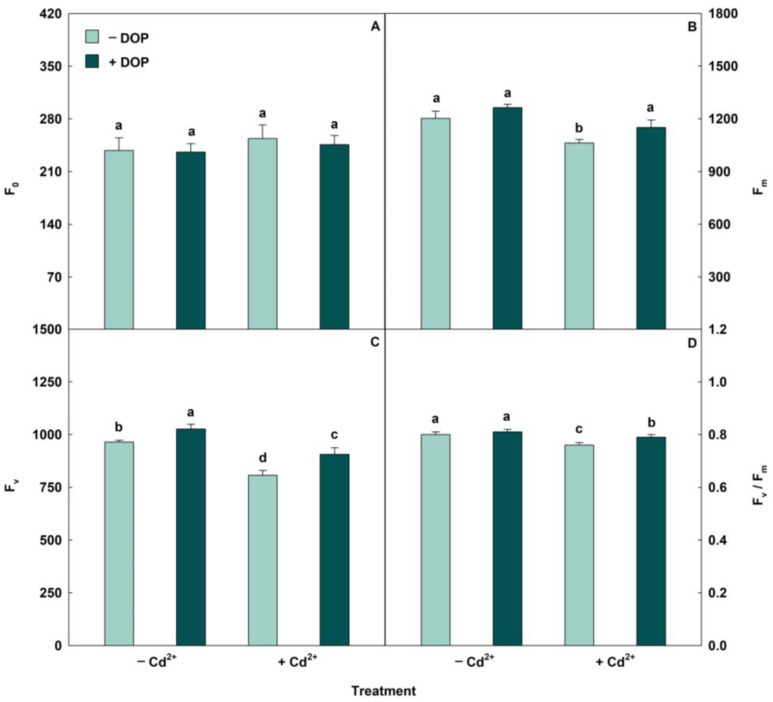
(**A**–**D**) Minimal fluorescence yield of the dark-adapted state (F_0_), maximal fluorescence yield of the dark-adapted state (F_m_), variable fluorescence (F_v_) and maximal quantum yield of PSII photochemistry (F_v_/F_m_) in soybean plants sprayed with DOP and exposed to Cd excess. Columns with different letters indicate significant differences from the Scott-Knott test (*p* < 0.05). Values described correspond to means from five repetitions and standard deviations.

**Figure 2 plants-14-03411-f002:**
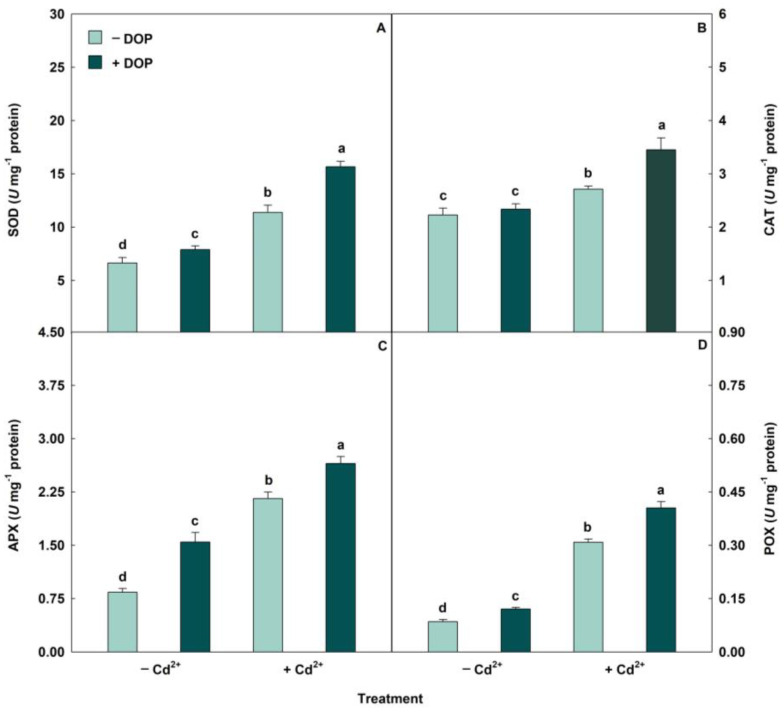
(**A**–**D**) Activities of superoxide dismutase (SOD), catalase (CAT), ascorbate peroxidase (APX) and peroxidase (POX) in soybean plants sprayed with DOP and Cd excess. Columns with different letters indicate significant differences from the Scott-Knott test (*p* < 0.05). Columns corresponding to means from five repetitions and standard deviations.

**Figure 3 plants-14-03411-f003:**
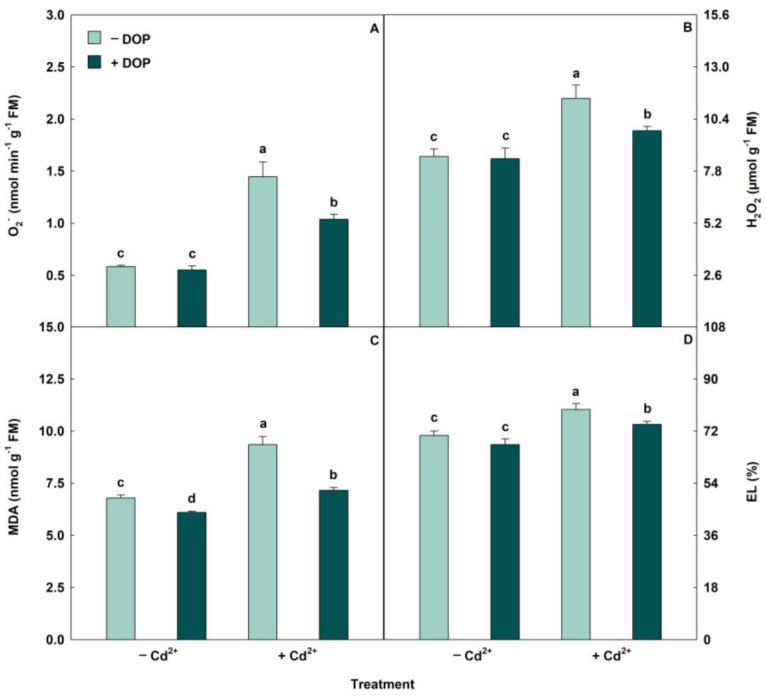
(**A**–**D**) Superoxide (O_2_^−^), hydrogen peroxide (H_2_O_2_), malondialdehyde (MDA) and electrolyte leakage (EL) in soybean plants sprayed with DOP and Cd excess. Columns with different letters indicate significant differences from the Scott-Knott test (*p* < 0.05). Columns corresponding to means from five repetitions and standard deviations.

**Figure 4 plants-14-03411-f004:**
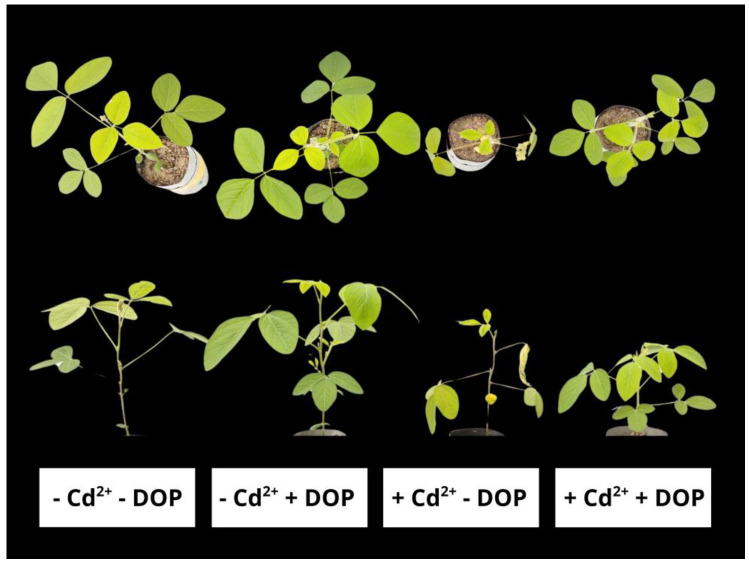
Upper and side views of soybean plants sprayed with DOP and Cd excess.

**Figure 5 plants-14-03411-f005:**
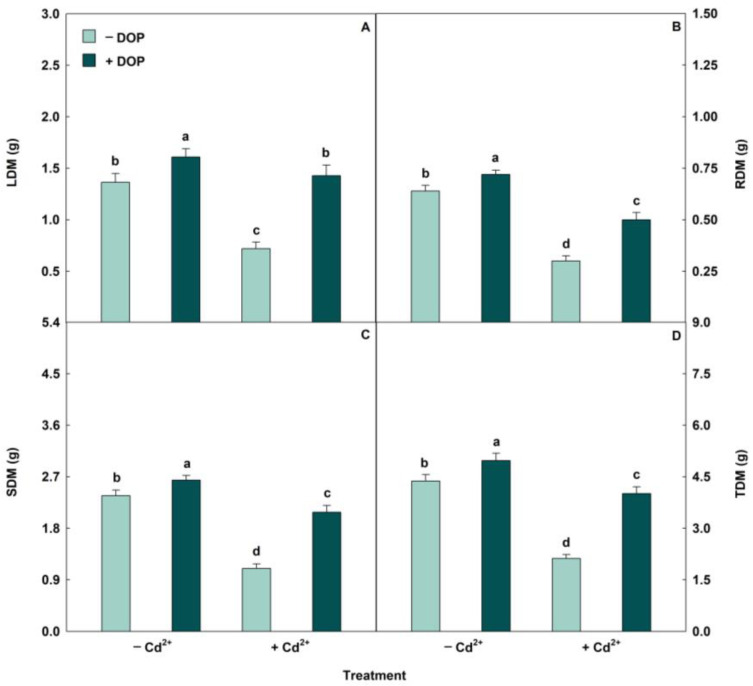
(**A**–**D**) Leaf dry matter (LDM), root dry matter (RDM), stem dry matter (SDM) and total dry matter (TDM) in soybean plants sprayed with DOP and Cd excess. Columns with different letters indicate significant differences from the Scott-Knott test (*p* < 0.05). Columns corresponding to means from five repetitions and standard deviations.

**Table 1 plants-14-03411-t001:** Cd contents in soybean plants sprayed with DOP and exposed to Cd excess.

Cd^2+^	DOP	Cd in Root (µg g DM^−1^)	Cd in Stem (µg g DM^−1^)	Cd in Leaf (µg g DM^−1^)
−	−	0.77 ± 0.05 c	0.09 ± 0.01 c	0.07 ± 0.01 c
−	+	0.75 ± 0.01 c	0.09 ± 0.01 c	0.06 ± 0.01 c
+	−	971.24 ± 24.31 a	5.05 ± 0.10 a	12.94 ± 0.20 a
+	+	387.26 ± 16.56 b	3.60 ± 0.07 b	8.01 ± 0.23 b

Cd^2+^ = cadmium; DOP = dopamine. Columns with different letters indicate significant differences from the Scott-Knott test (*p* < 0.05). Values described correspond to means from five repetitions and standard deviations.

**Table 2 plants-14-03411-t002:** Nutrient contents in soybean plants sprayed with DOP and exposed to Cd excess.

Cd^2+^	DOP	K (mg g DM^−1^)	Ca (mg g DM^−1^)	Mg (mg g DM^−1^)	Mn (µg g DM^−1^)	Fe (µg g DM^−1^)	Zn (µg g DM^−1^)
Contents in root							
−	−	34.52 ± 0.68 a	13.20 ± 0.60 a	14.05 ± 0.50 a	444.66 ± 21.76 a	4869.19 ± 156.89 a	50.75 ± 0.72 b
−	+	35.21 ± 0.35 a	13.36 ± 0.05 a	14.08 ± 0.20 a	449.99 ± 13.20 a	4907.15 ± 34.44 a	52.74 ± 0.81 a
+	−	23.54 ± 0.74 c	8.61 ± 0.49 c	10.61 ± 0.80 c	222.51 ± 5.78 c	2489.83 ± 106.04 c	28.21 ± 1.15 d
+	+	31.49 ± 0.92 b	11.80 ± 0.67 b	12.92 ± 0.32 b	352.31 ± 9.28 b	3901.83 ± 107.38 b	35.66 ± 1.21 c
Contents in stem							
−	−	61.06 ± 0.58 a	12.02 ± 0.49 b	2.55 ± 0.15 a	29.24 ± 0.40 a	44.07 ± 1.06 a	12.31 ± 1.03 a
−	+	62.00 ± 1.83 a	12.88 ± 0.26 a	2.65 ± 0.09 a	30.07 ± 0.34 a	45.41 ± 2.67 a	12.75 ± 0.55 a
+	−	41.64 ± 0.88 c	7.98 ± 0.11 d	1.26 ± 0.02 c	14.77 ± 0.53 c	24.23 ± 1.23 c	8.30 ± 0.63 c
+	+	49.75 ± 0.91 b	9.86 ± 0.64 c	1.90 ± 0.08 b	20.19 ± 1.09 b	32.59 ± 1.07 b	9.77 ± 0.66 b
Contents in leaf							
−	−	39.38 ± 0.57 a	14.02 ± 0.53 a	4.46 ± 0.13 a	74.17 ± 0.66 a	141.72 ± 1.44 a	25.87 ± 0.55 a
−	+	40.55 ± 1.40 a	14.51 ± 0.46 a	4.51 ± 0.10 a	75.35 ± 2.48 a	147.74 ± 1.76 a	26.75 ± 0.90 a
+	−	29.50 ± 1.13 c	10.51 ± 0.61 c	3.27 ± 0.15 c	56.46 ± 1.33 c	105.14 ± 3.47 c	20.48 ± 0.41 c
+	+	35.37 ± 0.26 b	12.22 ± 0.89 b	4.08 ± 0.08 b	69.18 ± 0.83 b	128.56 ± 7.74 b	22.96 ± 0.83 b

Cd^2+^ = cadmium; DOP = dopamine; K = potassium; Ca = calcium; Mg = magnesium; Mn = manganese; Fe = iron; Zn = zinc. Columns with different letters indicate significant differences from the Scott-Knott test (*p* < 0.05). Values described correspond to means from five repetitions and standard deviations.

**Table 3 plants-14-03411-t003:** Photosynthetic pigments, chlorophyll fluorescence and gas exchange in soybean plants sprayed with DOP and exposed to Cd excess.

Cd^2+^	DOP	Chl *a* (mg g^−1^ FM)	Chl *b* (mg g^−1^ FM)	Total Chl (mg g^−1^ FM)	Car (mg g^−1^ FM)	Ratio Chl *a*/Chl *b*	Ratio Total Chl/Car
−	−	12.00 ± 0.35 a	4.95 ± 0.29 a	16.94 ± 0.41 b	1.13 ± 0.06 a	2.43 ± 0.18 a	15.04 ± 0.98 a
−	+	12.30 ± 0.21 a	5.11 ± 0.14 a	17.42 ± 0.10 a	1.16 ± 0.03 a	2.41 ± 0.11 a	14.98 ± 0.51 a
+	−	6.95 ± 0.45 c	2.59 ± 0.19 c	9.54 ± 0.53 d	0.60 ± 0.03 c	2.69 ± 0.22 a	15.79 ± 0.60 a
+	+	9.50 ± 0.26 b	3.75 ± 0.26 b	13.25 ± 0.15 c	0.87 ± 0.02 b	2.54 ± 0.23 a	15.25 ± 0.45 a
Cd^2+^	DOP	Φ_PSII_	q_P_	NPQ	ETR (µmol m^−2^ s^−1^)	EXC (µmol m^−2^ s^−1^)	ETR/*P*_N_
−	−	0.381 ± 0.020 b	0.559 ± 0.025 a	0.90 ± 0.11 b	55.98 ± 2.89 b	0.525 ± 0.025 b	2.96 ± 0.20 a
−	+	0.405 ± 0.007 a	0.572 ± 0.010 a	0.80 ± 0.07 b	59.46 ± 1.07 a	0.502 ± 0.010 c	3.12 ± 0.02 a
+	−	0.285 ± 0.011 d	0.472 ± 0.013 b	1.10 ± 0.06 a	41.85 ± 1.69 d	0.626 ± 0.011 a	3.13 ± 0.09 a
+	+	0.358 ± 0.017 c	0.556 ± 0.036 a	1.03 ± 0.09 a	52.62 ± 2.57 c	0.545 ± 0.021 b	3.06 ± 0.23 a
Cd^2+^	DOP	*P*_N_ (µmol m^−2^ s^−1^)	*E* (mmol m^−2^ s^−1^)	*g*_s_ (mol m^−2^ s^−1^)	*C*_i_ (µmol mol^−1^)	WUE (µmol mmol^−1^)	*P*_N_/*C*_i_ (µmol m^−2^ s^−1^ Pa^−1^)
−	−	18.91 ± 0.56 a	1.86 ± 0.07 b	0.228 ± 0.011 a	232 ± 7 b	10.15 ± 0.25 b	0.082 ± 0.004 b
−	+	19.05 ± 0.40 a	1.70 ± 0.07 c	0.246 ± 0.021 a	201 ± 14 c	11.24 ± 0.60 a	0.095 ± 0.008 a
+	−	13.38 ± 0.79 c	2.03 ± 0.03 a	0.132 ± 0.013 c	262 ± 23 a	6.60 ± 0.43 d	0.051 ± 0.005 d
+	+	17.31 ± 0.93 b	1.90 ± 0.03 b	0.178 ± 0.013 b	245 ± 8 b	9.12 ± 0.44 c	0.071 ± 0.003 c

Cd^2+^ = cadmium; DOP = dopamine; Chl *a* = chlorophyll a; Chl *b* = chlorophyll b; Total chl = total chlorophyll; Car = carotenoids; Φ_PSII_ = effective quantum yield of PSII photochemistry; q_P_ = photochemical quenching coefficient; NPQ = nonphotochemical quenching; ETR = electron transport rate; EXC = relative energy excess at the PSII level; ETR/*P*_N_ = ratio between the electron transport rate and net photosynthetic rate; *P*_N_ = net photosynthetic rate; *E* = transpiration rate; *g*_s_ = stomatal conductance; *C*_i_ = intercellular CO_2_ concentration; WUE = water-use efficiency; *P*_N_/*C*_i_ = carboxylation instantaneous efficiency. Columns with different letters indicate significant differences from the Scott-Knott test (*p* < 0.05). Values described correspond to means from five repetitions and standard deviations.

## Data Availability

Data are available upon request to the corresponding author.

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
