# Peer review of "Dopamine Spraying Protects Against Cadmium-Induced Oxidative Stress and Stimulates Photosynthesis in Soybean Plants"

_plants, 2025, doi:10.3390/plants14223411_

Round 1

Reviewer 1 Report

Comments and Suggestions for Authors

This study demonstrates that exogenous DOP significantly enhances soybean tolerance to cadmium (Cd) stress by regulating antioxidant systems, improving nutrient uptake, and protecting photosynthetic mechanisms. The core mechanisms include reducing Cd accumulation, activating SOD activity, and restoring photosystem II functionality. The research provides new strategies for ecological restoration of heavy metal-contaminated farmland. As a low-cost and easily applied plant growth regulator, DOP could be widely promoted in crops like soybeans to ensure food safety. However, I have the following questions and suggestions regarding this article.

Question 1: How many soybean varieties were used for this study? Why was this varietie used for the test?Is the response of this variety to cadmium representative in soybean cultivation?

Question 2: Was a preliminary experiment conducted to determine the optimal Cd treatment concentration? What is the basis for selecting 500 µ M Cd as the treatment level? Does this concentration have representativeness or production significance

Question 3: The determination of physiological indicators related to the article is not comprehensive, and representative indicators such as malondialdehyde content, conductivity, and proline content are not clearly defined. It is also unclear through what pathway DOP improves soybean's cadmium resistance.

Question 4: Why were Fo, Fm, and Fv chosen for measurement? Is there any literature reporting the relationship between these three parameters and chlorophyll? Please describe and cite in Results 2.3.

Question 5: In Results 2.3, after Cd treatment, Fm and Fv showed significant decreases, while Fo remained unchanged. What is the reason for this? Please describe in Results 2.3.

Question 6: In the Materials and Methods section, all four experimental groups were measured for photosynthetic pigments, gas exchange, ROS content, and biomass at 30 days. Why were 30-day-old plants selected? Please provide supporting references.

Question 7: Line 12, "stimulating" to "by stimulating."

Question 8: Line 40, "environmental adversities" to "environmental stresses."

Question 9: Line 42, "have a low tolerance to" to "exhibit low tolerance to."

Question 10: Line103, “treatment with 100 µM DOP in plants stressed by Cd2+” to “treatment with 100 μM DOP in plants under Cd²⁺stress”.

Question 11: Line 107, Fo, Fm, and Fv appear for the first time, their full names should be clearly stated. 

Question 12: Lines 107 and 109, F0 should be changed to Fo.

Question 13: Line 116, “plants stressed with Cd2+ when sprayed with DOP obtained increases in PN, gs, WUE, and PN/Ci of 29%, 35%, 38%, and 39%”to “when plants under Cd²⁺stress were sprayed with DOP, there were increases of 29%, 35%, 38%, and 39% in PN, gs, WUE, and PN/Ci”.

Author Response

Dear reviewers and editor linked to manuscript #plants-3886970:

We are submitting the revised manuscript to the journal submission site for your review.

As instructed in your e-mail on 23 Set 2025, we have carefully considered all the reviewer’s comments and fully addressed them in the revised manuscript. Our responses to each specific reviewer comments are as follows (red into manuscript):

Decision from Editor: Your manuscript has been reviewed by experts in the field and we request that you make major revisions before it is processed further. Please revise your manuscript according to the reviewers' comments and upload the revised file within 10 days. Please click on the "Peer Review Reports" below to find the reviewers' comments and the version of your manuscript to be used for your revisions.

Decision: Major revision

Reviewer 1 Comments: This study demonstrates that exogenous DOP significantly enhances soybean tolerance to cadmium (Cd) stress by regulating antioxidant systems, improving nutrient uptake, and protecting photosynthetic mechanisms. The core mechanisms include reducing Cd accumulation, activating SOD activity, and restoring photosystem II functionality. The research provides new strategies for ecological restoration of heavy metal-contaminated farmland. As a low-cost and easily applied plant growth regulator, DOP could be widely promoted in crops like soybeans to ensure food safety. However, I have the following questions and suggestions regarding this article.

Reviewer 1: Question 1: How many soybean varieties were used for this study? Why was this varietie used for the test?Is the response of this variety to cadmium representative in soybean cultivation?

Authors: The plant material was included in manuscript “Seeds of Glycine max (L.) Merr. var. M8644IPRO Monsoy™”. This variety is often used in agricultural areas in Brazil.

Reviewer 1: Question 2: Was a preliminary experiment conducted to determine the optimal Cd treatment concentration? What is the basis for selecting 500 µ M Cd as the treatment level? Does this concentration have representativeness or production significance.

Authors: This information was included “To simulate Cd2+ exposure, CdCl2 was used at concentrations of 0 and 500 μM Cd, defined as this Cd concentration in agreement with Santos et al. (2018) and applied over 10 days (day 20–30 after the start of the experiment), with a 5-day interval between applications.”

Reviewer 1: Question 3: The determination of physiological indicators related to the article is not comprehensive, and representative indicators such as malondialdehyde content, conductivity, and proline content are not clearly defined. It is also unclear through what pathway DOP improves soybean's cadmium resistance.

Authors: These responses can be found in the abstract (below):

Cadmium (Cd) is a non-essential element that induces reactive oxygen species (ROS) production and damages the photosynthetic apparatus. Dopamine (DOP) is a neurotransmitter that plays an antioxidant role in metabolism. This research aimed to investigate exogenous DOP mitigates Cd-induced oxidative stress in soybean by assessing antioxidant metabolism, stress indicators, nutritional status, pigments, chlorophyll fluorescence, gas exchange, and biomass. The experiment was randomized with four treatments: two with Cd concentrations (0 and 500 µM Cd, described as – Cd and + Cd, respectively) and two DOP levels (0 and 100 µM DOP described as – DOP and + DOP, respectively). DOP mitigated Cd-induced damage by enhancing the antioxidant system and protecting the photosynthetic apparatus. This neurotransmitter positively modulated the enzymes superoxide dismutase (38%), catalase (27%), ascorbate peroxidase (23%), and peroxidase (31%), alleviating Cd-induced oxidative stress. In addition, DOP promoted increases in the effective quantum yield of PSII photochemistry (26%), photochemical quenching coefficient (18%), and electron transport rate (26%). Simultaneously, the neurotransmitter stimulated increases in the net photosynthetic rate (29%), stomatal conductance (35%), water use efficiency (38%), and instantaneous carboxylation efficiency (39%). Our results indicate that DOP exogenous increases tolerance to Cd-induced stress in soybean plants.

Reviewer 1: Question 4: Why were Fo, Fm, and Fv chosen for measurement? Is there any literature reporting the relationship between these three parameters and chlorophyll? Please describe and cite in Results 2.3.

Authors: Photosynthetic pigments, including chlorophylls, are responsible for absorbing light energy for photosynthesis. Additionally, Fo, Fm, and Fv are essential variables linked to chlorophyll fluorescence, being a process by which excited chlorophyll molecules re-emit some of this absorbed light energy as a faint red or far-red glow. This emitted fluorescence serves as a non-invasive tool to monitor plant health and photosynthetic activity.

              Additional information and literature were included in the discussion (below):

“The application of DOP attenuated the toxic effects caused by excess Cd on chlorophyll fluorescence. The reduction in the F0 value and the increase in Fv, Fm, and Fv/Fm indicate the improvements induced by the neurotransmitter for the PSII reaction center, improving its ability to collect and convert light energy. Cd stress impairs the absorption and conversion capacity of pigment molecules within PSII reaction centers [71]. On the other hand, DOP enhances the activity and stability of photosystems, particularly PSII, and also increases the concentration of antenna proteins [72]. Our results demonstrate that the changes found in Fv originate from Fm. In other words, Cd stress inhibited Fm. Fm is a key parameter in chlorophyll fluorescence in plants, being the highest level of fluorescence emitted by chlorophyll in a leaf after a plant is subjected to a dark-adapted state. The difference between Fm and F0 yields the Fv, which is then used in the ratio Fv/Fm, a common indicator of the plant's photosynthetic efficiency.

Decreases in Fv/Fm were observed in common bean plants exposed to Cd [73]. Piriformospora indica on sunflower plants under different levels of Cd contamination revealed that under high Cd concentration, the parameters Fm and Fv/Fm were reduced, while there was an increase in F0 [74]. Similarly, Malus hupehensis leaves treated with 100 μM DOP showed an increase in Fv/Fm [75].”

Reviewer 1: Question 5: In Results 2.3, after Cd treatment, Fm and Fv showed significant decreases, while Fo remained unchanged. What is the reason for this? Please describe in Results 2.3.

Authors: This information was inserted in the manuscript (discussion). “Our results demonstrate that the changes found in Fv originate from Fm. In other words, Cd stress inhibited Fm. Fm is a key parameter in chlorophyll fluorescence in plants, being the highest level of fluorescence emitted by chlorophyll in a leaf after a plant is subjected to a dark-adapted state. The difference between Fm and F0 yields the Fv, which is then used in the ratio Fv/Fm, a common indicator of the plant's photosynthetic efficiency.”

Reviewer 1: Question 6: In the Materials and Methods section, all four experimental groups were measured for photosynthetic pigments, gas exchange, ROS content, and biomass at 30 days. Why were 30-day-old plants selected? Please provide supporting references.

Authors: Based on preliminary tests, it was necessary to wait 30 days to carry out physiological and morphological, nutritional, and biochemical analyses, and Cd excess exercise mainly affects morphological variables.

Reviewer 1: Question 7: Line 12, "stimulating" to "by stimulating."

Authors: Change implemented in manuscript.

Reviewer 1: Question 8: Line 40, "environmental adversities" to "environmental stresses."

Authors: Change implemented in manuscript.

Reviewer 1: Question 9: Line 42, "have a low tolerance to" to "exhibit low tolerance to."

Authors: Change implemented in manuscript.

Reviewer 1: Question 10: Line103, “treatment with 100 µM DOP in plants stressed by Cd2+” to “treatment with 100 μM DOP in plants under Cd²⁺stress”.

Reviewer 1: Question 11: Line 107, Fo, Fm, and Fv appear for the first time, their full names should be clearly stated.

Authors: Change implemented in manuscript.

Reviewer 1: Question 12: Lines 107 and 109, F0 should be changed to Fo.

Authors: Thank you, but the correct term for this variable is F₀ (with ‘zero’) to represent the minimal chlorophyll fluorescence in the dark-adapted state.

Reviewer 1: Question 13: Line 116, “plants stressed with Cd2+ when sprayed with DOP obtained increases in PN, gs, WUE, and PN/Ci of 29%, 35%, 38%, and 39%”to “when plants under Cd²⁺stress were sprayed with DOP, there were increases of 29%, 35%, 38%, and 39% in PN, gs, WUE, and PN/Ci”.

Authors: Change implemented in manuscript.

Reviewer 2 Comments: The manuscript “Dopamine spraying protects against cadmium-induced oxidative stress and stimulates photosynthesis in soybean plants” presents a timely and interesting study on the role of exogenous dopamine (DOP) in mitigating cadmium (Cd) toxicity in soybean. The work is relevant because heavy metal contamination is a global agricultural issue, and dopamine is emerging as a promising bioactive molecule in plant stress physiology. The results are comprehensive, covering antioxidant metabolism, nutrient uptake, chlorophyll pigments, fluorescence, gas exchange, and biomass, which provides a broad mechanistic view of dopamine’s protective role. However, several aspects need clarification and improvement before the manuscript can be accepted.

The main strengths of the paper are the well-structured experimental design, the combination of multiple physiological and biochemical parameters, and the consistency of results showing that dopamine application alleviates Cd-induced oxidative stress and enhances photosynthesis. The data are generally convincing and support the conclusions.

Reviewer 2: Nonetheless, some major issues should be addressed. First, the novelty of this study should be better emphasized in the introduction and discussion. Although dopamine has been tested under various abiotic stresses, including heavy metals, the authors should more clearly explain what is new about their findings in soybean under Cd toxicity, compared to prior studies on Phaseolus vulgaris, Eucalyptus, and Malus. Second, the molecular mechanisms remain speculative. The discussion mentions gene expression (e.g., MdTyDC), but no transcript or protein data were generated in this study. The authors should explicitly state this limitation and suggest future directions involving transcriptomic or proteomic validation. Third, the concentrations of Cd (500 µM) and DOP (100 µM) used are relatively high. The ecological or agronomic relevance of these doses should be discussed, and ideally a dose–response experiment would be valuable.

Authors: Information linked to novelty of this study (introduction), future directions (conclusion) were inserted in the manuscript (below). Research involving dose-response experiments should be implemented in future studies. Thank you for the suggestion.

“Considering that the literature does not provide results on DOP roles in soybean plants under Cd stress and recurrent environmental contamination by heavy metals, this study aimed to investigate whether the exogenous application of DOP can mitigate oxidative stress and the repercussions on the photosynthetic apparatus in soybean plants under Cd intoxication, evaluating responses associated with antioxidant metabolism, stress indicators, nutritional status, photosynthetic pigments, chlorophyll fluorescence, gas exchange, and biomass.”

“Our results indicate that DOP exogenously increases tolerance to Cd-induced stress in soybean plants, but future studies involving transcriptomic or proteomic validation are necessary.”

Reviewer 2: Additional clarifications are also necessary. The methods section should describe in greater detail how dopamine was applied (volume per plant, spraying uniformity, possible solvent effects). It would also be useful to include visual evidence (photos of plants, chlorosis, or biomass reduction) in the results. Figures and tables should consistently report the number of replicates (n) and include precise statistical details (p-values, type of test used, post-hoc grouping). In the discussion, some statements about mechanisms (e.g., dopamine improving PSII stability or regulating nutrient transport) should be better supported by citations or toned down if only speculative.

Authors: More details on Materials and Methods were inserted in the manuscript. Figure 4 was elaborated and inserted into the manuscript, revealing visual evidence. Information related to the statistical details was included in the manuscript (Figures and tables). Discussion was improved with addition of recent references.

“12-day-old plants were sprayed with dopamine (DOP) or Milli-Q water (containing a proportion of ethanol equally used to prepare the DOP solution), spraying 10 mL per plant at 5-day intervals until day 30. The DOP solution (100 µM; Sigma–Aldrich, USA) was prepared as described previously [30].”

“Fig. 4. Upper and side views of soybean plants sprayed with DOP and Cd excess.”

“Columns with different letters indicate significant differences from the Scott-Knott test (P<0.05). Values described corresponding to means from five repetitions and standard deviations.”

“Cd toxicity damages the structures of chloroplasts and thylakoids, causing disturbances in the PSII reaction center, destruction of the oxygen-evolving complex (OEC), and blockage of electron transfer from QA to QB. These effects inhibit energy transfer and photochemical activity, resulting in a reduction of the photosynthetic rate [22,80,81].”

Reviewer 2: Minor comments include: the abstract is somewhat long and could be condensed by focusing on the most important numerical outcomes; the introduction contains redundant background information on Cd stress and could be shortened; language polishing is required in several places for clarity and conciseness; and the funding statement should be corrected according to MDPI requirements.

Authors: Modifications were implemented into the abstract, the introduction was improved, and the funding was adjusted (below):

“Cadmium (Cd) is a non-essential element that induces reactive oxygen species (ROS) production and damages the photosynthetic apparatus. Dopamine (DOP) is a neurotransmitter that plays an antioxidant role in metabolism. This research aimed to investigate exogenous DOP mitigates Cd-induced oxidative stress in soybean by assessing antioxidant metabolism, stress indicators, nutritional status, pigments, chlorophyll fluorescence, gas exchange, and biomass. The experiment was randomized with four treatments: two with Cd concentrations (0 and 500 µM Cd, described as – Cd and + Cd, respectively) and two DOP levels (0 and 100 µM DOP described as – DOP and + DOP, respectively). DOP mitigated Cd-induced damage by enhancing the antioxidant system and protecting the photosynthetic apparatus. This neurotransmitter positively modulated the enzymes superoxide dismutase (38%), catalase (27%), ascorbate peroxidase (23%), and peroxidase (31%), alleviating Cd-induced oxidative stress. In addition, DOP promoted increases in the effective quantum yield of PSII photochemistry (26%), photochemical quenching coefficient (18%), and electron transport rate (26%). Simultaneously, the neurotransmitter stimulated increases in the net photosynthetic rate (29%), stomatal conductance (35%), water use efficiency (38%), and instantaneous carboxylation efficiency (39%). Our results indicate that DOP exogenous increases tolerance to Cd-induced stress in soybean plants.”

“The root, the main organ for nutrient uptake, is also the first tissue affected by Cd stress, which reduces its growth and compromises the plant’s nutritional status [11,12] This effect arises from ionic competition at the root surface [13], where Cd²⁺ competes with essential elements, particularly divalent cations, for transport channels, leading to disturbances in nutrient assimilation and in the specific functions of these elements [14].”

“Funding: This research had financial support from Fundação Amazônia de Amparo a Estudos e Pesquisas (FAPESPA/Brazil), Conselho Nacional de Desenvolvimento Científico e Tecnológico (CNPq/Brazil) and Universidade Federal Rural da Amazônia (UFRA/Brazil) to AKSL. On the other hand, ASC was supported with a scholarship from Coordenação de Aperfeiçoamento de Pessoal de Nível Superior (CAPES/Brazil).”

Reviewer 2: In conclusion, the manuscript presents valuable results and demonstrates that exogenous dopamine alleviates Cd-induced oxidative stress and enhances photosynthetic performance in soybean. With improved emphasis on novelty, a clearer acknowledgment of limitations, and careful editing of the methods, discussion, and figures, this paper could make a solid contribution to the field of plant stress physiology. I recommend major revision before publication.

Authors: Careful revision of the methods, discussion, and figures was performed.

Reviewer 3 Comments: The manuscript deals with an important topic related to the role of dopamine spraying in the protection against cadmium-induced oxidative stress and its stimulation of photosynthesis in soybean plants. The manuscript technically sounds well and shows high novelty. However, it needs major linguistic adjustments; therefore, I invite the authors to pass their manuscript to a native English speaker for editing and revision. In this regard, the needed adjustments are highlighted in “Minor comments” section. Also, numerous statements are attributed to old sources (references) that should be replaced by more recent ones (last five years of publication).

The Abstract section outlines clearly the problematic, aims, methodology and findings of the current study while reporting the main conclusions aroused. The Introduction section is well structured and aiming and underlines appropriately the whole subject under study. The aims of the study are also clear and understood. The Materials and Methods section is clear, well written, and encloses all the information related to the adopted methodology, and statistical analysis. The Results section shows a correct statistical representation and an adequate scientific analysis of the findings. The Discussion section outlines an appropriate comparison of the present findings with previously published ones in literature. An appropriate Conclusions section was added in which authors summarized the findings of their study. However, they shall suggest further related research being based on the raised assumptions.

My comments and queries for authors are detailed below in “Major comments” and “Minor comments” sections.

Reviewer 3: The manuscript needs major linguistic adjustments; accordingly, I invite the authors to pass their manuscript to a native English speaker for editing and revision. Most needed adjustments are highlighted in “Minor comments” section.

Authors: Manuscript was totally revised again using language editing services by a traditional publisher.

Reviewer 3: The list of references shall be up-to-date (last five years of publication).

Authors: Current references were improved, using recent publications, when possible.

Reviewer 3: Conclusions: Authors shall suggest further related research being based on the raised assumptions from the current study.

Authors: Additional research suggested in the manuscriptour results indicate that DOP exogenously increases tolerance to Cd-induced stress in soybean plants, but future studies involving transcriptomic or proteomic validation are necessary.”

Reviewer 3: Abstract: Page 1, line 23: Kindly adjust as follow: “modulated”.

Authors: Change implemented in manuscript.

Reviewer 3: Introduction: Page 1, line 34: Kindly adjust as follow: “is one”.

Authors: Change implemented in manuscript.

Reviewer 3: Introduction: Page 1, line 38: Kindly adjust as follow: “In the 2023­­­­­-2024”.

Authors: Change implemented in manuscript.

Reviewer 3: Introduction: Page 2, lines ­44-46: “Cadmium… agriculture”: Reference 8 used for this statement is a little bit old (older than 5 previous years); accordingly, kindly replace it by the following recent and reliable one: “doi:10.1016/j.heliyon.2024.e27138”.

Authors: Change implemented in manuscript and suggested reference added.

Reviewer 3: Introduction: Page 2, lines 47-49: “The root… status”: The references used for these statements are old; accordingly, kindly replace them by more recent ones (last five years of publication).

Authors: The reference has been updated.

Reviewer 3: Introduction: Page 2, lines 47-53: “The root… the elements”: The sentence is long and cumbersome; accordingly, kindly reformulate in order to make it more concise, clearer, and more aiming.

Authors: Change implemented in manuscript. “The root, the main organ for nutrient uptake, is also the first tissue affected by Cd stress, which reduces its growth and compromises the plant’s nutritional status [11,12] This effect arises from ionic competition at the root surface [13], where Cd²⁺ competes with essential elements, particularly divalent cations, for transport channels, leading to disturbances in nutrient assimilation and in the specific functions of these elements [14].”

Reviewer 3: Introduction: Page 2, lines 59-61: “In physiological… Calvin cycle”: The reference used for this statement is a little bit old; accordingly, kindly replace it by a more recent one (last five years of publication).

Authors: The reference has been updated.

Reviewer 3: Introduction: Page 2, line 70: Kindly adjust the reference presentation form following the journal2s guidelines.

Authors: Refences adjusted following the journal’s guidelines

Reviewer 3: Introduction: Page 2, line 72: Reference 27 is lacking in the text!!

Authors: Change implemented in manuscript.

Reviewer 3: Introduction: Page 2, line 72: Reference 29 is old; accordingly, kindly replace it by a recent one (last five years of publication).

Authors: This modification was not possible to implement; however, our reference is from 2018 (less than 10 years).

Reviewer 3: Introduction: Page 2, lines 73-77: “Often… peroxidation”: The sentence is badly written in standard English; accordingly, kindly reformulate it.

Authors: Paragraph was reformulated It frequently enhances stress tolerance by positively regulating nutrient uptake, transport, and assimilation [36], improving gas exchange, photosynthetic pigments, and chlorophyll fluorescence parameters [37], as well as boosting antioxidant enzyme activities, reducing ROS accumulation [38] and protecting cells from lipid peroxidation [39].

Reviewer 3: Introduction: Page 2, lines ­78-79: “We hypothesize… machinery”: Kindly avoid the first voice form of the sentence and adopt the impersonal form instead.

Authors: Change implemented in manuscript.

Reviewer 3: Results, 2.3. DOP alleviated the Cd impacts on photosynthetic apparatus: Page 3, lines 116-119: “However… Cd2+”: The sentence is badly written in standard English; accordingly, kindly reformulate it.

Authors: Reformulated sentence However, Dopamine-treated plants exposed to excess Cd promoted increases of 26%, 18%, and 26% in ΦPSII, qP, and ETR, respectively, and reductions of 6%, 13%, and 2% in NPQ, EXC, and ETR/PN, respectively, compared with plants exposed to the same Cd treatment without dopamine.”

Reviewer 3: Results, 2.5. Neurotransmitter reduced the harmful effects caused by Cd excess in biomass: Page 3, line 129: Kindly adopt a more suitable title for this paragraph.

Authors: Subtitle was reformulated “DOP mitigated impacts caused by Cd excess on biomass”

Reviewer 3: Results, 2.5. Neurotransmitter reduced the harmful effects caused by Cd excess in biomass: Page 3, lines 131-133: “However… excess”: The sentence is badly written in standard English; accordingly, kindly reformulate it.

Authors: Paragraph was reformulated “Biomass was significantly reduced in plants exposed to Cd excess (Fig. 4). However, DOP sprayed in plants stressed with Cd2+ induced increases in LDM, RDM, SDM, and TDM of 89%, 67%, 98%, and 89%, respectively, compared to plants exposed only to Cd excess.”

Reviewer 3: Discussion: Page 4, lines 138-141: “DOP… to shoot”: The sentence is cumbersome; accordingly, kindly reformulate in order to make it clearer and more aiming.

Authors: Reformulated sentence “Dopamine alleviates Cd-induced stress by regulating the expression of genes involved in metal uptake and detoxification. In particular, overexpression of the MdTyDC gene, associated with neurotransmitter synthesis, helps reduce Cd²⁺ transport to the shoots”

Reviewer 3: Discussion: Page 4, lines 141-142: Kindly replace “Like our study” by “Similarly”.

Authors: Change implemented in manuscript.

Reviewer 3: Discussion: Page 4, line 142: Kindly adjust the presentation form of the reference following the journal’s guidelines.

Authors: Refences adjusted following the journal’s guidelines

Reviewer 3: Discussion: Page 4, line 146: Reference 65 is a little bit old; accordingly, kindly replace it by a more recent one (last five years of publication).

Authors: The reference has been updated.

Reviewer 3: Discussion: Page 4, line 147: Kindly adjust the presentation form of the reference following the journal’s guidelines.

Authors: Refences adjusted following the journal’s guidelines

Reviewer 3: Discussion: Page 4, lines 147-149: “Similar… stem”: The sentence is badly written in standard English; accordingly, kindly reformulate it.

Authors: Change implemented in manuscript “High Cd levels in Eucalyptus urophylla plants promoted immobilization of this metal in the roots, reducing Cd levels in leaves and stems.”

Reviewer 3: Discussion: Page 4, lines 155-156: “Additionally… rate”: The references used for this statement are old; accordingly, kindly replace them by more recent ones (last five years of publication).

Authors: The reference has been updated.

Reviewer 3: Discussion: Page 4, lines 156-159: “Cd2+… permeability”: Reference 69 is old; accordingly, kindly replace it by a more recent one (last five years of publication).

Authors: The reference has been updated.

Reviewer 3: Discussion: Page 4, lines 159-162: “The Cd… to roots”: The sentence is badly written in standard English; accordingly, kindly reformulate it.

Authors: Reformulated sentence “Cd²⁺ uptake occurs through transporters of essential mono- and divalent metals, competing with cations such as K⁺, Ca²⁺, Mg²⁺, Mn²⁺, Fe²⁺, and Zn²⁺, as well as through ion transport channels, thereby reducing the uptake of essential elements by the roots.”

Reviewer 3: Discussion: Page 4, line 165: Kindly adjust the presentation form of the reference following the journal’s guidelines.

Authors: Refences adjusted following the journal’s guidelines

Reviewer 3: Discussion: Page 4, line 167: Kindly adjust as follow: “alleviated”.

Authors: Change implemented in manuscript.

Reviewer 3: Discussion: Page 4, lines 168-169: Kindly adjust the reference presentation form of the reference following the journal’s guidelines.

Authors: Refences adjusted following the journal’s guidelines.

Reviewer 3: Discussion: Page 4, lines 179 and 183: Same recommendation as in the previous comment.

Authors: Refences adjusted following the journal’s guidelines.

Reviewer 3: Discussion: Page 4, line 182: Kindly adjust as follow: “compared to”.

Authors: Change implemented in manuscript.

Reviewer 3: Discussion: Page 4, lines 188-189: “DOP… capacity”: The reference used for this statement is old; accordingly, kindly replace it by a recent one (last five years of publication).

Authors: The reference has been updated.

Reviewer 3: Discussion: Page 5, lines 194-196: Kindly adjust the presentation form of the references following the journal’s guidelines.

Authors: Refences adjusted following the journal’s guidelines

Reviewer 3: Discussion: Page 5, lines 198-200: “These results… by Cd”: The sentence is badly written in standard English; accordingly, kindly reformulate it.

Authors: Reformulated sentence “These results corroborate our observations, demonstrating that the increase in antioxidant enzyme activities induced by DOP attenuates the oxidative stress caused by Cd.”

Reviewer 3: Discussion: Page 5, lines 206-210: “Cd stress… complex”: The references used for this statement are old; accordingly, kindly replace them by recent ones (last five years).

Authors: The reference has been updated.

Reviewer 3: Discussion: Page 5, lines 210-214: “Furthermore… photosystem II”: Reference 84 is a little bit old; accordingly, kindly replace it by a more recent one (last five years of publication).

Authors: This modification was not possible to implement; however, our reference is from 2018 (less than 10 years).

Reviewer 3: Discussion: Page 5, lines 217 and 219: Kindly adjust the presentation form of the references following the journal’s guidelines.

Authors: Refences adjusted following the journal’s guidelines

Reviewer 3: Discussion: Page 5, lines 229-230: Kindly adjust as follow: “On the other hand”.

Authors: Change implemented in manuscript.

Reviewer 3: Discussion: Page 5, lines 229-231: “On the other hand… antenna proteins”: The sentence is badly written in standard English; accordingly, kindly reformulate it.

Authors: Reformulated sentence “On the other hand, DOP enhances the activity and stability of photosystems, particularly PSII, and also increases the concentration of antenna proteins.”

Reviewer 3: Discussion: Page 5, lines 233-236: Kindly adjust the presentation form of the references following the journal’s guidelines.

Authors: Refences adjusted following the journal’s guidelines.

Reviewer 3: Discussion: Page 5, lines 235-237: “Similar… DOP”: Kindly avoid the first voice form of the sentence and adopt the impersonal form instead.

Authors: Change implemented in manuscript.

Reviewer 3: Discussion: Page 6, lines 246-248: “Low… apparatus”: The reference used for this statement is old; accordingly, kindly replace it by a recent one (last five years of publication).

Authors: The reference has been updated.

Reviewer 3: Discussion: Page 6, lines 248 and 252: Kindly adjust the presentation form of the references following the journal’s guidelines.

Authors: Refences adjusted following the journal’s guidelines.

Reviewer 3: Discussion: Page 6, lines 251-256: “Additionally… overexpression”: The sentence is cumbersome; accordingly, kindly reformulate in order to make it clearer and more aiming.

Authors: Reformulated sentence “Additionally, the role of the TyDC gene in DOP biosynthesis in apple plants and its contribution to drought stress response resulted in increases in Fv/Fm, ΦPSII, and qp, as well as a reduction in NPQ in transgenic lines (overexpressing MdTyDC) after 40 days of stress. These effects were associated with the higher DOP accumulation derived from MdTyDC overexpression [79].”

Reviewer 3: Discussion: Page 6, line 257: Kindly adjust as follow: “Pre-treatment”.

Authors: Change implemented in manuscript.

Reviewer 3: Discussion: Page 6, line 260: Kindly adjust as follow: “and reducing”.

Authors: Change implemented in manuscript.

Reviewer 3: Discussion: Page 6, lines 258-264: “Cd toxicity… assimilation”: References 97 and 98 are old; accordingly, kindly replace them by recent ones (last five years of publication).

Authors: Updated references in the manuscript and reformulated sentence “Cd toxicity damages the structures of chloroplasts and thylakoids, causing disturbances in the PSII reaction center, destruction of the oxygen-evolving complex (OEC), and blockage of electron transfer from QA to QB. These effects inhibit energy transfer and photochemical activity, resulting in a reduction of the photosynthetic rate [22,80,81].”

Reviewer 3: Discussion: Page 6, lines 265-267: “The transpiration… in E”: The reference used for this statement is a little bit old; accordingly, kindly replace it by a more recent one (last five years of publication).

Authors: The reference has been updated

Reviewer 3: Discussion: Page 6, lines 267-269: “Reductions… (2018)”: The sentence is a little bit cumbersome; accordingly, kindly reformulate in order to make it clearer and more aiming.

Authors: Reformulated sentence “Cowpea plants exposed to Cd exhibited reductions in PN, gs, E, WUE, and PN/Ci, ac-companied by an increase in Ci [85].”

Reviewer 3: Discussion: Page 6, line 269: Kindly adjust the presentation form of the references following the journal’s guidelines.

Authors: Refences adjusted following the journal’s guidelines.

Reviewer 3: Discussion: Page 6, line 269: Kindly adjust as follow: “pre-treatment”.

Authors: Change implemented in manuscript.

Reviewer 3: Discussion: Page 6, lines 269-272: “Lan… DOP”: The sentence is badly written in standard English; accordingly, kindly reformulate it.

Authors: Reformulated sentence Pre-treatment with different concentrations of DOP (0, 50, 100, 150, and 200 μM) in cucumber plants exposed to nitrate-induced stress (50 or 500 μM) markedly increased PN, gs, E, and Ci following DOP application [37].”

Reviewer 3: Discussion: Page 6, lines 280-282: Kindly adjust the presentation form of these references following the journal’s guidelines.

Authors: Refences adjusted following the journal’s guidelines.

Reviewer 3: Materials and Methods, 4.2. Plants, containers, and acclimation: Page 6, line 295: Same recommendation as in the previous comment.

Authors: Refences adjusted following the journal’s guidelines.

Reviewer 3: Materials and Methods, 4.4. Dopamine (DOP) preparation and application: Page 7, line 309: Same recommendation as in the previous two comments.

Authors: Refences adjusted following the journal’s guidelines.

Reviewer 3: Materials and Methods, 4.5. Plant nutrition and Cd excess: Page 7, lines 312 and 314: Same recommendation as in the previous comments.

Authors: Refences adjusted following the journal’s guidelines.

Reviewer 3: Materials and Methods, 4.6. Chlorophyll fluorescence and gaseous exchange: Page 7, lines 322 and 324: Same recommendation as in the previous comments.

Authors: Refences adjusted following the journal’s guidelines.

Reviewer 3: Materials and Methods, 4.7. Assessment of antioxidant enzymes, soluble proteins, and stress markers: Page 7, lines 328 and 330-332: Kindly adjust as follow: “quantified as”.

Authors: Change implemented in manuscript.

Reviewer 3: Materials and Methods, 4.7. Assessment of antioxidant enzymes, soluble proteins, and stress markers: Page 7, line 336: Kindly adjust the presentation form of the reference following the journal’s guidelines.

Authors: Refences adjusted following the journal’s guidelines.

Reviewer 3: Materials and Methods, 4.8. Assessment of photosynthetic pigments, nutritional composition, and biomass: Page 8, line 345: Same recommendation as in the previous comment.

Authors: Refences adjusted following the journal’s guidelines.

Reviewer 3: Conclusions: Page 8, line 359: Kindly remove “This research revealed that”.

Authors: Change implemented in manuscript.

Reviewer 3: Conclusions: Page 8, line 361: Kindly adjust as follow: “reduced”.

Authors: Change implemented in manuscript.

Reviewer 3: Conclusions: Page 8, lines 361-368: “Exogenous… exchange”: The sentence is long and cumbersome; accordingly, kindly reformulate in order to make it more concise, clearer, and more aiming.

Authors: Reformulated sentence “Exogenous DOP reduced oxidative damage by enhancing the activities of antioxidant enzymes, including superoxide dismutase, catalase, ascorbate peroxidase, and peroxidase, which detoxify reactive compounds such as hydrogen peroxide, superoxide, malondialdehyde, and electrolytes. Consequently, DOP alleviated negative effects on the photosynthetic apparatus, improving chlorophyll content, photosynthetic activity, PSII efficiency, and gas exchange in plants under Cd stress.”

Reviewer 3: Conclusions: Page 8, line 372: Kindly add a sentence at the end of this section in which you suggest further related research being based on the raised assumptions from the current study.

Authors: Suggested further researchOur results indicate that DOP exogenously increases tolerance to Cd-induced stress in soybean plants, but future studies involving transcriptomic or proteomic validation are necessary.”

We thank you again for your time and effort in handling and reviewing our manuscript and we are looking forward to hearing from you.

Sincerely yours,

Allan Klynger da Silva Lobato

Professor / Universidade Federal Rural da Amazônia

Affiliate member / Brazilian Academy of Sciences

+55 91 993134006

allan.lobato@ufra.edu.br

Reviewer 2 Report

Comments and Suggestions for Authors

The manuscript “Dopamine spraying protects against cadmium-induced oxidative stress and stimulates photosynthesis in soybean plants” presents a timely and interesting study on the role of exogenous dopamine (DOP) in mitigating cadmium (Cd) toxicity in soybean. The work is relevant because heavy metal contamination is a global agricultural issue, and dopamine is emerging as a promising bioactive molecule in plant stress physiology. The results are comprehensive, covering antioxidant metabolism, nutrient uptake, chlorophyll pigments, fluorescence, gas exchange, and biomass, which provides a broad mechanistic view of dopamine’s protective role. However, several aspects need clarification and improvement before the manuscript can be accepted.

The main strengths of the paper are the well-structured experimental design, the combination of multiple physiological and biochemical parameters, and the consistency of results showing that dopamine application alleviates Cd-induced oxidative stress and enhances photosynthesis. The data are generally convincing and support the conclusions.

Nonetheless, some major issues should be addressed. First, the novelty of this study should be better emphasized in the introduction and discussion. Although dopamine has been tested under various abiotic stresses, including heavy metals, the authors should more clearly explain what is new about their findings in soybean under Cd toxicity, compared to prior studies on Phaseolus vulgaris, Eucalyptus, and Malus. Second, the molecular mechanisms remain speculative. The discussion mentions gene expression (e.g., MdTyDC), but no transcript or protein data were generated in this study. The authors should explicitly state this limitation and suggest future directions involving transcriptomic or proteomic validation. Third, the concentrations of Cd (500 µM) and DOP (100 µM) used are relatively high. The ecological or agronomic relevance of these doses should be discussed, and ideally a dose–response experiment would be valuable.

Additional clarifications are also necessary. The methods section should describe in greater detail how dopamine was applied (volume per plant, spraying uniformity, possible solvent effects). It would also be useful to include visual evidence (photos of plants, chlorosis, or biomass reduction) in the results. Figures and tables should consistently report the number of replicates (n) and include precise statistical details (p-values, type of test used, post-hoc grouping). In the discussion, some statements about mechanisms (e.g., dopamine improving PSII stability or regulating nutrient transport) should be better supported by citations or toned down if only speculative.

Minor comments include: the abstract is somewhat long and could be condensed by focusing on the most important numerical outcomes; the introduction contains redundant background information on Cd stress and could be shortened; language polishing is required in several places for clarity and conciseness; and the funding statement should be corrected according to MDPI requirements.

In conclusion, the manuscript presents valuable results and demonstrates that exogenous dopamine alleviates Cd-induced oxidative stress and enhances photosynthetic performance in soybean. With improved emphasis on novelty, a clearer acknowledgment of limitations, and careful editing of the methods, discussion, and figures, this paper could make a solid contribution to the field of plant stress physiology. I recommend major revision before publication.

Author Response

(The authors gave the same response as above.)

Reviewer 3 Report

Comments and Suggestions for Authors

Comments to the Author:

Title: Dopamine spraying protects against cadmium-induced oxidative stress and stimulates photosynthesis in soybean plants

Overview and general recommendation:

The manuscript deals with an important topic related to the role of dopamine spraying in the protection against cadmium-induced oxidative stress and its stimulation of photosynthesis in soybean plants. The manuscript technically sounds well and shows high novelty. However, it needs major linguistic adjustments; therefore, I invite the authors to pass their manuscript to a native English speaker for editing and revision. In this regard, the needed adjustments are highlighted in “Minor comments” section. Also, numerous statements are attributed to old sources (references) that should be replaced by more recent ones (last five years of publication).

The Abstract section outlines clearly the problematic, aims, methodology and findings of the current study while reporting the main conclusions aroused. The Introduction section is well structured and aiming and underlines appropriately the whole subject under study. The aims of the study are also clear and understood. The Materials and Methods section is clear, well written, and encloses all the information related to the adopted methodology, and statistical analysis. The Results section shows a correct statistical representation and an adequate scientific analysis of the findings. The Discussion section outlines an appropriate comparison of the present findings with previously published ones in literature. An appropriate Conclusions section was added in which authors summarized the findings of their study. However, they shall suggest further related research being based on the raised assumptions.

My comments and queries for authors are detailed below in “Major comments” and “Minor comments” sections.

1- Major comments:

  1.  The manuscript needs major linguistic adjustments; accordingly, I invite the authors to pass their manuscript to a native English speaker for editing and revision. Most needed adjustments are highlighted in “Minor comments” section.
  2. The list of references shall be up-to-date (last five years of publication).
  3. Conclusions: Authors shall suggest further related research being based on the raised assumptions from the current study.

2- Minor comments:

  1. Abstract: Page 1, line 23: Kindly adjust as follow: “modulated”.
  2. Introduction: Page 1, line 34: Kindly adjust as follow: “is one”.
  3. Introduction: Page 1, line 38: Kindly adjust as follow: “In the 2023­­­­­-2024”.
  4. Introduction: Page 2, lines ­44-46: “Cadmium… agriculture”: Reference 8 used for this statement is a little bit old (older than 5 previous years); accordingly, kindly replace it by the following recent and reliable one: “doi:10.1016/j.heliyon.2024.e27138”.
  5. Introduction: Page 2, lines 47-49: “The root… status”: The references used for these statements are old; accordingly, kindly replace them by more recent ones (last five years of publication).
  6. Introduction: Page 2, lines 47-53: “The root… the elements”: The sentence is long and cumbersome; accordingly, kindly reformulate in order to make it more concise, clearer, and more aiming.
  7. Introduction: Page 2, lines 59-61: “In physiological… Calvin cycle”: The reference used for this statement is a little bit old; accordingly, kindly replace it by a more recent one (last five years of publication).
  8. Introduction: Page 2, line 70: Kindly adjust the reference presentation form following the journal2s guidelines.
  9. Introduction: Page 2, line 72: Reference 27 is lacking in the text!!
  10. Introduction: Page 2, line 72: Reference 29 is old; accordingly, kindly replace it by a recent one (last five years of publication).
  11. Introduction: Page 2, lines 73-77: “Often… peroxidation”: The sentence is badly written in standard English; accordingly, kindly reformulate it.
  12. Introduction: Page 2, lines ­78-79: “We hypothesize… machinery”: Kindly avoid the first voice form of the sentence and adopt the impersonal form instead.
  13. Results, 2.3. DOP alleviated the Cd impacts on photosynthetic apparatus: Page 3, lines 116-119: “However… Cd2+”: The sentence is badly written in standard English; accordingly, kindly reformulate it.
  14. Results, 2.5. Neurotransmitter reduced the harmful effects caused by Cd excess in biomass: Page 3, line 129: Kindly adopt a more suitable title for this paragraph.
  15. Results, 2.5. Neurotransmitter reduced the harmful effects caused by Cd excess in biomass: Page 3, lines 131-133: “However… excess”: The sentence is badly written in standard English; accordingly, kindly reformulate it.
  16. Discussion: Page 4, lines 138-141: “DOP… to shoot”: The sentence is cumbersome; accordingly, kindly reformulate in order to make it clearer and more aiming.
  17. Discussion: Page 4, lines 141-142: Kindly replace “Like our study” by “Similarly”.
  18. Discussion: Page 4, line 142: Kindly adjust the presentation form of the reference following the journal’s guidelines.
  19. Discussion: Page 4, line 146: Reference 65 is a little bit old; accordingly, kindly replace it by a more recent one (last five years of publication).
  20. Discussion: Page 4, line 147: Kindly adjust the presentation form of the reference following the journal’s guidelines.
  21. Discussion: Page 4, lines 147-149: “Similar… stem”: The sentence is badly written in standard English; accordingly, kindly reformulate it.
  22. Discussion: Page 4, lines 155-156: “Additionally… rate”: The references used for this statement are old; accordingly, kindly replace them by more recent ones (last five years of publication).
  23. Discussion: Page 4, lines 156-159: “Cd2+… permeability”: Reference 69 is old; accordingly, kindly replace it by a more recent one (last five years of publication).
  24. Discussion: Page 4, lines 159-162: “The Cd… to roots”: The sentence is badly written in standard English; accordingly, kindly reformulate it.
  25. Discussion: Page 4, line 165: Kindly adjust the presentation form of the reference following the journal’s guidelines.
  26. Discussion: Page 4, line 167: Kindly adjust as follow: “alleviated”.
  27. Discussion: Page 4, lines 168-169: Kindly adjust the reference presentation form of the reference following the journal’s guidelines.
  28. Discussion: Page 4, lines 179 and 183: Same recommendation as in the previous comment.
  29. Discussion: Page 4, line 182: Kindly adjust as follow: “compared to”.
  30. Discussion: Page 4, lines 188-189: “DOP… capacity”: The reference used for this statement is old; accordingly, kindly replace it by a recent one (last five years of publication).
  31. Discussion: Page 5, lines 194-196: Kindly adjust the presentation form of the references following the journal’s guidelines.
  32. Discussion: Page 5, lines 198-200: “These results… by Cd”: The sentence is badly written in standard English; accordingly, kindly reformulate it.
  33. Discussion: Page 5, lines 206-210: “Cd stress… complex”: The references used for this statement are old; accordingly, kindly replace them by recent ones (last five years).
  34. Discussion: Page 5, lines 210-214: “Furthermore… photosystem II”: Reference 84 is a little bit old; accordingly, kindly replace it by a more recent one (last five years of publication).
  35. Discussion: Page 5, lines 217 and 219: Kindly adjust the presentation form of the references following the journal’s guidelines.
  36. Discussion: Page 5, lines 229-230: Kindly adjust as follow: “On the other hand”.
  37. Discussion: Page 5, lines 229-231: “On the other hand… antenna proteins”: The sentence is badly written in standard English; accordingly, kindly reformulate it.
  38. Discussion: Page 5, lines 233-236: Kindly adjust the presentation form of the references following the journal’s guidelines.
  39. Discussion: Page 5, lines 235-237: “Similar… DOP”: Kindly avoid the first voice form of the sentence and adopt the impersonal form instead.
  40. Discussion: Page 6, lines 246-248: “Low… apparatus”: The reference used for this statement is old; accordingly, kindly replace it by a recent one (last five years of publication).
  41. Discussion: Page 6, lines 248 and 252: Kindly adjust the presentation form of the references following the journal’s guidelines.
  42. Discussion: Page 6, lines 251-256: “Additionally… overexpression”: The sentence is cumbersome; accordingly, kindly reformulate in order to make it clearer and more aiming.
  43. Discussion: Page 6, line 257: Kindly adjust as follow: “Pre-treatment”.
  44. Discussion: Page 6, line 260: Kindly adjust as follow: “and reducing”.
  45. Discussion: Page 6, lines 258-264: “Cd toxicity… assimilation”: References 97 and 98 are old; accordingly, kindly replace them by recent ones (last five years of publication).
  46. Discussion: Page 6, lines 265-267: “The transpiration… in E”: The reference used for this statement is a little bit old; accordingly, kindly replace it by a more recent one (last five years of publication).
  47. Discussion: Page 6, lines 267-269: “Reductions… (2018)”: The sentence is a little bit cumbersome; accordingly, kindly reformulate in order to make it clearer and more aiming.
  48. Discussion: Page 6, line 269: Kindly adjust the presentation form of the references following the journal’s guidelines.
  49. Discussion: Page 6, line 269: Kindly adjust as follow: “pre-treatment”.
  50. Discussion: Page 6, lines 269-272: “Lan… DOP”: The sentence is badly written in standard English; accordingly, kindly reformulate it.
  51. Discussion: Page 6, lines 280-282: Kindly adjust the presentation form of these references following the journal’s guidelines.
  52. Materials and Methods, 4.2. Plants, containers, and acclimation: Page 6, line 295: Same recommendation as in the previous comment.
  53. Materials and Methods, 4.4. Dopamine (DOP) preparation and application: Page 7, line 309: Same recommendation as in the previous two comments.
  54. Materials and Methods, 4.5. Plant nutrition and Cd excess: Page 7, lines 312 and 314: Same recommendation as in the previous comments.
  55. Materials and Methods, 4.6. Chlorophyll fluorescence and gaseous exchange: Page 7, lines 322 and 324: Same recommendation as in the previous comments.
  56. Materials and Methods, 4.7. Assessment of antioxidant enzymes, soluble proteins, and stress markers: Page 7, lines 328 and 330-332: Kindly adjust as follow: “quantified as”.
  57. Materials and Methods, 4.7. Assessment of antioxidant enzymes, soluble proteins, and stress markers: Page 7, line 336: Kindly adjust the presentation form of the reference following the journal’s guidelines.
  58. Materials and Methods, 4.8. Assessment of photosynthetic pigments, nutritional composition, and biomass: Page 8, line 345: Same recommendation as in the previous comment.
  59. Conclusions: Page 8, line 359: Kindly remove “This research revealed that”.
  60. Conclusions: Page 8, line 361: Kindly adjust as follow: “reduced”.
  61. Conclusions: Page 8, lines 361-368: “Exogenous… exchange”: The sentence is long and cumbersome; accordingly, kindly reformulate in order to make it more concise, clearer, and more aiming.
  62. Conclusions: Page 8, line 372: Kindly add a sentence at the end of this section in which you suggest further related research being based on the raised assumptions from the current study.
Comments on the Quality of English Language

The manuscript needs major linguistic adjustments; accordingly, I invite the authors to pass their manuscript to a native English speaker for editing and revision. Most needed adjustments are highlighted in “Minor comments” section.

Author Response

(The authors gave the same response as above.)

Round 2

Reviewer 1 Report

Comments and Suggestions for Authors

Currently, genetically modified soybeans are widely planted in Brazil, and the varieties mentioned in the author's article are not conventional varieties widely used in Brazil. Considering the genotype differences between soybean varieties, it is recommended that the author supplement physiological measurement experiments on other soybean varieties (preferably including recipient varieties of genetically modified soybeans) treated with cadmium by applying DOP externally. In addition, it is necessary to compare the physiological test indicators of DOP applied at different concentration gradients to determine the optimal concentration of DOP applied in production.

Author Response

Dear reviewers and editor linked to manuscript # plants-3886970:

We are submitting the revised manuscript to the journal submission site for your review.

As instructed in your e-mail on 11 Ot 2025, we have carefully considered all the reviewer’s comments and fully addressed them in the revised manuscript. Our responses to each specific reviewer comments are as follows (red into manuscript):

Reviewer 1 Comments: Currently, genetically modified soybeans are widely planted in Brazil, and the varieties mentioned in the author's article are not conventional varieties widely used in Brazil. Considering the genotype differences between soybean varieties, it is recommended that the author supplement physiological measurement experiments on other soybean varieties (preferably including recipient varieties of genetically modified soybeans) treated with cadmium by applying DOP externally. In addition, it is necessary to compare the physiological test indicators of DOP applied at different concentration gradients to determine the optimal concentration of DOP applied in production.

Authors: We respect the opinion, experience and time dedicated by reviewer 1 in reviewing this manuscript, but we would like to express our opinion so that the editor-in-chief can make the better possible decision.

We disagree that the variety used is not/was not widely used in Brazil. The variety M8644IPRO Monsoy™ was the most commonly used in the North region of Brazil for 12 years (2010 to 2022). Currently, this material is used in plant breeding programs for North region from Brazil, as it has broad adaptability.

This manuscript used only one variety of soybean, but future research may be conducted with other plant materials (other manuscript).

Our research presented 54 variables, with 30 specifically being physiological/biochemical variables (55%). In future research, we may consider other variables.

The DOP concentration used in this research was considered based on the available literature and preliminary tests.

Reviewer 2 Report

Comments and Suggestions for Authors

The authors have implemented the suggested changes and addressed the comments provided in the review. The text has been revised both in terms of content and editorial quality. In its current form, the article meets the requirements set by the journal and raises no objections.

Author Response

Dear reviewers and editor linked to manuscript # plants-3886970:

We are submitting the revised manuscript to the journal submission site for your review.

As instructed in your e-mail on 11 Oct 2025, we have carefully considered all the reviewer’s comments and fully addressed them in the revised manuscript. Our responses to each specific reviewer comments are as follows (red into manuscript):

Reviewer 2 Comments: The authors have implemented the suggested changes and addressed the comments provided in the review. The text has been revised both in terms of content and editorial quality. In its current form, the article meets the requirements set by the journal and raises no objections.

Authors: Thank you very much.

Reviewer 3 Report

Comments and Suggestions for Authors

Comments to the Author:

Title: Dopamine spraying protects against cadmium-induced oxidative stress and stimulates photosynthesis in soybean plants

Overview and general recommendation:

Authors have made huge improvements to their manuscript and are well thanked for that. Only minor adjustments are still needed mainly related to the brief explanation of provided methodologies in the Materials and Methods section.

My comments and queries for authors are detailed below in “Minor comments” section.

Minor comments:

  1. Discussion: Page 5, lines 192-194: “Similarly… [63]”: The sentence is badly written in standard English; accordingly, kindly reformulate it.
  2. Discussion: Page 5, line 215: Kindly adjust as follow: “contents”.
  3. Discussion: Page 5, lines 232-235: “Piriformospora… [74]”: The sentence is badly written in standard English; accordingly, kindly reformulate it.
  4. Discussion: Page 6, line 269: Kindly adjust as follow: “Ci”.
  5. Materials and Methods, 4.4. Dopamine (DOP) preparation and application: Page 7, line 307: Kindly describe briefly the adopted methodology.
  6. Materials and Methods, 4.6. Chlorophyll fluorescence and gaseous exchange: Page 7, line320: Same recommendation as in the previous comment.
  7. Materials and Methods, 4.7. Assessment of antioxidant enzymes, soluble proteins, and stress markers: Page 7, line 332: Same recommendation as in the previous two comments.
  8. Materials and Methods, 4.8. Assessment of photosynthetic pigments, nutritional composition, and biomass: Page 7, line 339: Same recommendation as in the previous comments.
  9. Conclusions: Page 8, line 362-363: Kindly adjust as follow: “increased”.

Author Response

Dear reviewers and editor linked to manuscript # plants-3886970:

We are submitting the revised manuscript to the journal submission site for your review.

As instructed in your e-mail on 11 Oct 2025, we have carefully considered all the reviewer’s comments and fully addressed them in the revised manuscript. Our responses to each specific reviewer comments are as follows (red into manuscript):

Reviewer 3 Comments: Authors have made huge improvements to their manuscript and are well thanked for that. Only minor adjustments are still needed mainly related to the brief explanation of provided methodologies in the Materials and Methods section. My comments and queries for authors are detailed below in “Minor comments” section.

Authors: N/A

Reviewer 3: Discussion: Page 5, lines 192-194: “Similarly… [63]”: The sentence is badly written in standard English; accordingly, kindly reformulate it.

Authors: This sentence was improved “Exogenous treatment using 100 μM DOP in lettuce plants subjected to nitrogen deficiency mitigated oxidative stress, stimulating the antioxidant defense system [63].”

Reviewer 3: Discussion: Page 5, line 215: Kindly adjust as follow: “contents”.

Authors: Correction implemented.

Reviewer 3: Discussion: Page 5, lines 232-235: “Piriformospora… [74]”: The sentence is badly written in standard English; accordingly, kindly reformulate it.

Authors: This sentence was reformulated “Decreases in Fv/Fm were observed in common bean plants exposed to Cd [73]. Cd contamination in sunflower plants promoted reductions in Fm and Fv/Fm, while there was an increase in F0 [74]. In other hand, Malus hupehensis leaves treated with 100 μM DOP showed an increase in Fv/Fm [75].

Reviewer 3: Discussion: Page 6, line 269: Kindly adjust as follow: “Ci”.

Authors: Correction implemented.

Reviewer 3: Materials and Methods, 4.4. Dopamine (DOP) preparation and application: Page 7, line 307: Kindly describe briefly the adopted methodology.

Authors: This paragraph was improved (below):

“12-day-old plants were sprayed with dopamine (DOP) or Milli-Q water (containing a proportion of ethanol equally used to prepare the DOP solution), spraying 10 mL per plant at 5-day intervals until day 30. The DOP solution (100 µM; Sigma–Aldrich, USA) was prepared as described previously [30].”

Reviewer 3: Materials and Methods, 4.6. Chlorophyll fluorescence and gaseous exchange: Page 7, line320: Same recommendation as in the previous comment.

Authors: This paragraph was improved (below):

“Chlorophyll fluorescence was assessed using a modulated chlorophyll fluorometer (model OS5p; Opti-Sciences), being measured in fully expanded leaves under light. Preliminary tests determined that the acropetal third of leaves in the middle third of the plant and that adapted to the dark for 30 min yielded the greatest Fv/Fm ratio. Therefore, this part of the plant was used for measurements. The intensity and duration of the saturation light pulse were 7,500 µmol m–2.s–1 and 0.7 s, respectively [93]. Gas exchange was measured using an infrared gas analyzer (LCPro+; ADC BioScientific), being evaluated in all plants under a constant CO2 concentration (360 μmol mol-1 CO2), photo-synthetically active radiation (800 μmol photons m-2 s-1), air-flow rate (300 µmol s-1), and temperature (28 °C) in the test chamber between 10:00 and 12:00 h [92].”

Reviewer 3: Materials and Methods, 4.7. Assessment of antioxidant enzymes, soluble proteins, and stress markers: Page 7, line 332: Same recommendation as in the previous two comments.

Authors: This paragraph was improved (below):

“Antioxidant enzymes (SOD, CAT, APX, and POX) and soluble proteins were isolated from leaf tissues. The extraction mixture was prepared by homogenizing 500 mg of fresh plant material in 5 mL of extraction buffer, which consisted of 50 mM phosphate buffer (pH 7.6), 1.0 mM ascorbate, and 1.0 mM EDTA. Samples were centrifuged at 14,000 × g for 4 min at 3 °C, and the supernatant was collected [94]. Quantification of total soluble proteins was conducted [95]. SOD assay was performed at 560 nm [96], and SOD activity was quantified as units per mg of protein. CAT assay was measured at 240 nm [97], with CAT activity quantified as μmol H2O2 mg–1 protein min–1. The APX experiment was conducted at 290 nm [98], with APX activity quantified as μmol AsA mg–1 protein min–1. The POX assay was measured at 470 nm [99], with activity quantified as μmol tetraguaiacol mg–1 protein min–1. The O2 concentration was quantified at 530 nm [100]. Stress markers were isolated, specifically H2O2 and MDA [101]. The H2O2 concentration was quantified [102]. The MDA concentration was calculated utilizing an attenuation value of 155 mM-1 cm-1 [103]. EL was assessed using the methodology outlined in [104] and calculated as EL (%) = (EC1/EC2) × 100.”

Reviewer 3: Materials and Methods, 4.8. Assessment of photosynthetic pigments, nutritional composition, and biomass: Page 7, line 339: Same recommendation as in the previous comments.

Authors: This paragraph was improved (below):

“Chlorophyll and carotenoid concentrations were assessed using 40 mg of foliar tissue. The samples were homogenized in darkness using 8 mL of 90% methanol (Sig-ma-Aldrich™). The homogenate underwent centrifugation at 6000 × g for 10 minutes at 5ºC. The supernatant was discarded, and the concentrations of Chl a, Chl b, Car, and total Chl were measured using a spectrophotometer (model UV-M51; Bel Photonics) fol-lowing the protocol established in [105]. Samples (100 mg) of root, stem, and leaf tissues were pre-digested in 50 mL conical tubes using 2 mL of sub-boiled HNO3. Subsequently, 8 ml of a solution comprising 4 ml of H2O2 (30% v/v) and 4 mL of ultra-pure water was added and transferred to a Teflon digestion tube [106]. Cd, K, Ca, Mg, Mn, Fe, and Zn were quantified using an inductively coupled plasma mass spectrometer (model ICP-MS 7900; Agilent). The biomass of root, stem and leaves were quantified by constant dry weights (g) following desiccation in a forced-air oven at 65°C.”

Reviewer 3: Conclusions: Page 8, line 362-363: Kindly adjust as follow: “increased”.

Authors: Correction implemented.

We thank you again for your time and effort in handling and reviewing our manuscript and we are looking forward to hearing from you.

Sincerely yours,

Allan Klynger da Silva Lobato

Professor / Universidade Federal Rural da Amazônia

Affiliate member / Brazilian Academy of Sciences

+55 91 993134006

allan.lobato@ufra.edu.br

Round 3

Reviewer 1 Report

Comments and Suggestions for Authors

The author's response did not solve my problem. I suggest adding relevant experiments on other soybean varieties and experiments on other concentrations of DOP.

Author Response

Dear reviewers and editor linked to manuscript # plants-3886970:

We are submitting the revised manuscript to the journal submission site for your review.

As instructed in your e-mail on 25 Out 2025, we have carefully considered all the reviewer’s comments and fully addressed them in the revised manuscript. Our responses to each specific reviewer comments are as follows (red into manuscript):

Decision from Editor: Dear Authors,

Thank you very much for submitting the revised version of your manuscript and for carefully addressing most of the reviewers’ comments. We appreciate your efforts in improving the clarity and quality of the paper. However, we kindly ask you to improve the following additional points:

Decision: Minor revision

Editor: 1. Please revise the manuscript in accordance with the new minor comments provided by Reviewer 3.

Authors: All corrections requested by Reviewer 3 were implemented (two rounds).

Editor: 2. As highlighted by Reviewers 1 and 2, please emphasize more clearly the innovative aspects and originality of your research in the revised version. This will help strengthen the scientific contribution of your manuscript.

Authors: This information was included in the hypothesis “Considering that the literature does not provide results on DOP roles in soybean plants under Cd stress and recurrent environmental contamination by heavy metals, this study aimed to investigate whether the exogenous application of DOP can mitigate oxidative stress and the repercussions on the photosynthetic apparatus in soybean plants under Cd intoxication, evaluating responses associated with antioxidant metabolism, stress indicators, nutritional status, photosynthetic pigments, chlorophyll fluorescence, gas exchange, and biomass”.

Editor: 3. Please clarify the rationale for the selected doses of metal and dopamine as suggested by Reviewer 2, so that readers can better understand the experimental setup.

Authors: This information was included “DOP concentrations were chosen based on previous research [37]. Cd treatments were defined in preliminary tests and results available in literature [89].”

Editor: 4. We suggest merging Sections 4.3 and 4.5 into a single section to avoid repetition and to make the experimental design clearer and more coherent.

Authors: These sections were merged (below):

4.3. Experimental design, plant nutrition and Cd excess

The experiment was randomized with four treatments: two with cadmium concentrations (0 and 500 µM Cd, described as – Cd and + Cd, respectively) and two concentrations of dopamine (0 and 100 µM DOP described as – DOP and + DOP, respectively). Five replicates for each of the four treatments were conducted, yielding 20 experimental units in the experiment, with one plant in each experimental unit. DOP concentrations were chosen based on previous research [37]. Cd treatments were defined in preliminary tests and results available in literature [89].

Plants received the following macro- and micronutrients supplied in the nutrient solution as described previously [92]. To simulate Cd2+ exposure, cadmium chloride (CdCl2) was used at concentrations of 0 and 500 μM Cd and applied over 10 days (day 20–30 after the start of the experiment), with a 5-day interval between applications. During the study, the nutrient solutions were changed at 07:00 h at 3-day intervals, with the pH adjusted to 6.5 using HCl or NaOH. On day 30 of the experiment, all plants' physiological and morphological parameters were measured, and the leaf tissues were collected for nutritional and biochemical analyses.

Editor: 5. Following Reviewer’s 1suggestion, you may consider adding as a future research perspectives (e.g. in the ‘Conclusions’ section) a note on the need to examine other soybean varieties and to conduct experiments with different dopamine concentrations.

Authors: This information was included in conclusion section “but future studies with other soybean varieties and varying DOP concentrations, combined with transcriptomic or proteomic validation, are necessary.”

Reviewer 1: The author's response did not solve my problem. I suggest adding relevant experiments on other soybean varieties and experiments on other concentrations of DOP.

Authors: This information was included in conclusion section “but future studies with other soybean varieties and varying DOP concentrations, combined with transcriptomic or proteomic validation, are necessary.”

We thank you again for your time and effort in handling and reviewing our manuscript and we are looking forward to hearing from you.

Sincerely yours,

Allan Klynger da Silva Lobato

Professor / Universidade Federal Rural da Amazônia

Affiliate member / Brazilian Academy of Sciences

+55 91 993134006

allan.lobato@ufra.edu.br